# Evaluating the utility of amino acid similarity-aware kmers to represent TCR repertoires for classification

Hannah Kockelbergh[1,2], Shelley C. Evans[3,4], Liam Brierley[1,5], Peter L. Green[6], Andrea L. Jorgensen[1], Elizabeth J. Soilleux[3], Anna Fowler[1]*

**1** Department of Health Data Science, University of Liverpool, Liverpool, United Kingdom, **2** The Kennedy Institute of Rheumatology, University of Oxford, Oxford, United Kingdom, **3** Department of Pathology, University of Cambridge, Cambridge, United Kingdom, **4** Early Cancer Institute, University of Cambridge, Cambridge, United Kingdom, **5** MRC-University of Glasgow Centre for Virus Research, University of Glasgow, Glasgow, United Kingdom, **6** School of Engineering, University of Liverpool, Liverpool, United Kingdom

\* a.fowler@liverpool.ac.uk

## Abstract

Insights gained through interpretation of models trained on the T-cell receptor (TCR) repertoire contribute to advances in understanding of immune-mediated disease. This has the potential to improve diagnostic tests and treatments, particularly for autoimmune diseases. However, TCR repertoire datasets with samples from donors of known autoimmune disease status generally include orders of magnitude fewer samples than TCR sequences. Promising TCR repertoire classification approaches consider relationships between non-identical TCR sequences. In particular, kmer methods demonstrate strong and stable performance for small datasets. We propose a TCR repertoire representation that considers the relationships between amino acids within kmers flexibly and efficiently. XGBoost and logistic regression models are trained and tested on kmer representations of TCR repertoire datasets including samples from patients with coeliac disease as well as donors with previous cytomegalovirus infection. XGBoost models outperform logistic regression, indicating that interactions may be crucial for discriminative ability. We find that a reduced alphabet based on BLOSUM62 can lead to a model with slightly stronger XGBoost testing performance than other kmer features. Though it remains unclear whether there is an amino acid encoding that can substantially improve TCR repertoire classification with reduced alphabet kmers, evidence that this representation enables faster training of XGBoost models in comparison to kmer clusters suggests that our reduced alphabet approach permits wider exploration of amino acid similarity in practice. Finally, we detail motifs which are important in each top-performing XGBoost model and compare them to TCR sequences previously associated with each immune status. We highlight the challenge of interpreting non-linear TCR repertoire classification models

**Data availability statement:** Both coeliac disease training and testing datasets are available at ImmPort (https://www.immport.org) under Study Accession SDY2976. Coeliac disease associated TCR sequences were made available in the supplementary materials of Han et al. PNAS 2013 (doi:10.1073/pnas.1311861110). Cytomegalovirus datasets are made publicly available by the original authors vis Adaptive Biotechnologies ImmuneAccess at https://clients.adaptivebiotech.com/pub/emerson-2017-natgen. Cytomegalovirus-specific TCR sequences were downloaded from VDJdb https://vdjdb.cdr3.net accessed 11/10/25. Code used to generate TCR repertoire representations and perform classification, are available at https://github.com/hannrko/enc_kmer_tcr_models.

**Funding:** This work was funded by Coeliac UK and Innovate UK (INOV01-18 to EJS). H.K. received a Doctoral Training Studentship from EPSRC. The funders had no role in study design, data collection and analysis, decision to publish, or preparation of the manuscript.

**Competing interests:** The authors have declared that no competing interests exist.

trained on kmers which, if overcome, could lead to biomarker discovery for autoimmune diseases.

## Author summary

TCR repertoire classification models can provide valuable understanding of autoimmune diseases if they can accurately infer autoimmune disease status and are biologically interpretable. Based on a kmer representation of the TCR repertoire, which has been shown to be most appropriate to train classification models on smaller datasets out of three popular approaches, we develop a computationally efficient method of grouping amino acid sequences to add knowledge to immune status classification model inputs. We find that most of the 4mer-based feature types we tested perform well in combination with an XGBoost model, and that applying a halved alphabet of amino acids based on BLOSUM62 may be beneficial or neutral for immune status classification performance. We also consider the effect on models and features on interpretability, and conclude that although some insights may be gained from inspecting feature importance, dedicated explanatory methods are required to truly understand the complex relationships between kmers that are captured by our best-performing XGBoost models. While standard kmer XGBoost models have the shortest training time, our proposed reduced alphabet methodology presents a more efficient alternative to kmer clustering. Future exploration of amino acid similarity with encodings other than those based on Atchley factors or BLOSUM62, as well as length of kmers k, would benefit from our reduced alphabet representation over clustering of kmers.

## Introduction

T cells bind to specific antigens via their T-cell receptor (TCR). A naive T cell may clonally expand on encounter of an antigen to which its TCR specifically binds, resulting in a clonal population of T cells expressing the same TCR. TCRs may be formed of an $\alpha$ and $\beta$ chain encoded by the TRA and TRB loci, or $\gamma$ and $\delta$ chain encoded by the TRG and TRD loci; T cells expressing a TCR of each type, respectively, are called $\alpha\beta$ T cells or $\gamma\delta$ T cells, respectively. The diversity of TCRs expressed by an individual's T cells, called their TCR repertoire, is typically vast; it has been estimated that a TCR repertoire of a single individual may include 100 million unique TCR $\beta$ chains [1]. This diversity is underpinned by somatic recombination of Variable (V), Diversity (D) (only for $\beta$ and $\delta$ chains) and Joining (J) gene segments that form the TCR genes, with addition of a Constant gene segment [2]. The complementarity-determining region 3 (CDR3) at the junction of joined V, D and J gene segments is the most variable region of TCR-encoding genes, and is generally considered to be the most important contributor to TCR-antigen binding specificity [3]. Next generation sequencing has enabled vast libraries of CDR3 sequences encoding TCRs from many T cells to be profiled in bulk.

Comparison of TCR repertoire samples from individuals with and without an immune status of interest can provide improved understanding of immune-mediated conditions. This approach has been applied to viral infection [4–25], including a study of TCR repertoires in the context of cytomegalovirus (CMV) infection by Emerson et al. [4]. CMV infection is highly prevalent and generally asymptomatic. Though CMV infection is important to study due to clinical relevance in cases of organ transplantation, pregnancy and immunodeficiency, many studies utilise the Emerson et al. dataset to evaluate immune status classification models [7,8,10], likely due to its large size and inclusion of 2 separate cohorts.

Autoimmune disease has also been studied using the TCR repertoire [13,14,26–38]. Coeliac disease (CeD) is an underdiagnosed autoimmune disease triggered by dietary gluten, where CD4+ $\alpha\beta$ T cells bind to gluten-derived peptides presented by certain HLA-DQ molecules, resulting in damage to the small intestine [39]. Diagnosis of CeD may require a confirmatory duodenal biopsy, but valid findings rely on adequate consumption of gluten for a period beforehand. In addition, probability that two specialist pathologists disagree on the classification of a biopsy is estimated to be at least 20% [40], indicating that this test is somewhat subjective. $\gamma\delta$ T cell populations in gut tissue of CeD patients are different compared to those who do not have CeD [41], suggesting relevance of $\gamma\delta$ T cells in disease. Further understanding of CeD, as could be obtained through interpretation of a TCR repertoire classification model, may lead to new diagnostic tests.

While some "public" TCR sequences, which are TCR sequences that are observed in more than one individual, have been identified that are associated with immune status [4,5,22,24,27], consideration of shared attributes of TCRs within and between individuals may enable non-identical CDR3 sequences associated with a condition to be more readily identified, especially with smaller sample sizes where "public" TCRs are less likely to be observed. Strategies including deep learning [6,8], TCR clustering [15,42] and approaches that split TCR sequences into subsequences of length k, or kmers [7,43–46] have been proposed which enable TCR similarity to be considered. Deep learning methods such as attention-based networks [8] or convolutional neural networks [6] are capable of capturing complex relationships, and enable the whole length of the CDR3 sequence to be used as input to a model, which may preserve the context of amino acids within it. However, kmer methods have been shown to perform just as well for small datasets, and exhibit more stable performance [7]. In addition, an internal benchmark on real TCR data shows that a kmer-based support vector machine performs almost as well as DeepRC [8]. In this work, kmers are targeted for further development due to their suitability as a classification basis for high dimensional TCR repertoire datasets with a small number of samples.

Within the CDR3 region of a TCR, it has been proposed that similar amino acids may lead to shared binding capabilities [43,46]. It follows that incorporating amino acid relationships, defined by their physicochemical properties or likelihood of substitution for one-another, within a kmer representation may lead to recognition of non-identical kmers that share the same function within a TCR repertoire. However, if that is the case, the underlying relationships between amino acids that may lead to shared function are not well-established, although arginine (R), proline (P) and glycine (G) have been found to be informative for TCR specificity across $\alpha$ and $\beta$ chains [47]. Previously, Atchley factors [48] have been utilised to encode kmers [43–46,49,50] in two related approaches, which both represent a kmer using a vector, where 5 Atchley factors are concatenated for each amino acid in the kmer. Models are then trained on encoded kmers [43–45,50] or clusters of encoded kmers [46]. A study which benchmarks these 2 immune status classification approaches suggests that they have comparable performance to each other, but are both surpassed by a kmer-based gradient-boosted tree model that does not consider amino acid properties [7]. Though it is suggested that amino acid property information may improve the performance of their gradient-boosted tree model, it remains unclear to what extent the kmer representation or classification model of choice is responsible for differences in classification performance [7].

In kmeans clustering as undertaken by Thomas et al., the number of clusters is specified in advance. 10 and 100 clusters are tried in their original publication [46], which each imply different amino acid similarity thresholds. Since training time would be expected to scale in proportion to the number of clusters [51], rigorously assessing a wide range of number of clusters, and therefore kmer similarity thresholds, is expected to be rather time-consuming. The study by Katayama and Kobayashi also highlights long training time of over a week for the method by Ostmeyer et al., despite their final model

being limited to capturing linear relationships between the kmer representation and immune status [7]. An kmer-based representation was therefore developed which enables efficient assessment of multiple amino acid similarity thresholds.

This work compares kmer representations of the TCR repertoire for immune status classification to explore the utility of including amino acid similarity information. A novel representation of the TCR repertoire based on kmers defined with a reduced alphabet is proposed, in which similar amino acids are deemed to be equivalent. Our approach enables efficient and flexible exploration of TCR repertoire representations, allowing a wide range of amino acid relationships to be considered within an immune status classification model. This TCR representation method is benchmarked against the kmer clustering by Thomas et al. and unaltered kmers with no encoding applied. Additional results are also presented with amino acid similarity defined using BLOSUM62 [52], which to our knowledge is the first use of BLOSUM in a kmer representation of the TCR repertoire. These representations are used to train XGBoost and logistic regression models for CMV infection status and CeD status which are each evaluated using separate testing datasets, wherein methodology is otherwise kept consistent for fair comparison. As a result, evidence is provided to show the effect of amino acid similarity information on TCR repertoire classification performance with a kmer representation, which may guide future methodological development for TCR repertoire classification for small datasets.

## Results

### A flexible definition of amino acid reduced alphabets

Atchley factors and BLOSUM62 are used to define relationships between the 20 naturally-occurring amino acids. Reduced alphabets are defined by considering amino acids equivalent if they share similarity, where amino acid similarity is determined by similarity of Atchley factors or substitution likelihood by BLOSUM62. In practice, any user-defined amino acid properties or substitution matrix could be employed. Hierarchical clustering of amino acids results in multiple solutions which are defined at different similarity thresholds, which allows size of the reduced alphabet to be tuned as a hyperparameter. When applied to a kmer representation of the TCR repertoire, this flexibly-defined reduced alphabet enables optimal amino acid relationships to be chosen for a TCR repertoire classification model.

For each of Atchley factors and BLOSUM62, reduced alphabets over a full range of similarity thresholds, and therefore of size 1–20, are shown in Fig 1. For Atchley factors, 20 different clustering solutions exist over a range of distances. For BLOSUM62, 19 different clustering solutions exist due to immediate transition from 15 to 17 clusters at distance of 0.333: equivalent pairwise similarity is calculated between glutamic acid (E) and glutamine (Q), as well as lysine (K) and arginine (R). Each of these reduced alphabets are applied to a 4mer representation of each dataset.

In contrast, kmer clusters defined similarly as by Thomas et al. [46] define groups of whole kmers rather than amino acids. As in the original method, we use Atchley factors to calculate kmer clusters, and additionally BLOSUM62, with 100 clusters in each case. This fixed number of 100 clusters is used here since it is found to result in a more sensitive and specific TCR repertoire classification model than with 10 clusters in the original publication [46]. Also, fitting the number of clusters as a hyperparameter over a full range of possible values is expected to lead to a substantial computational challenge due to the average computational complexity of kmeans clustering scaling linearly with the number of clusters [51]. With the idea that clusters of kmers are ultimately grouped together due to an underlying amino acid similarity threshold, reduced alphabet size may be compared to number of clusters in kmer clustering. Any reduced alphabet of size 3 results in 81 groups of 4mers, and that of size 4 results in 256 groups, therefore an alphabet of size 3 is the most comparable to 100 kmer clusters. The amino acid similarity threshold implied by each of these should be roughly equivalent if the number of 4mers observed and clustered with kmeans is near the full observable set of 4mers, of which there are $20^4 = 160,000$. We define our standard kmer representation to be 4mers defined with an amino alphabet alphabet size of 20, similar to that by Katayama and Kobayashi [7], the only differences being their use of 3mers instead of 4mers, as well as end characters.

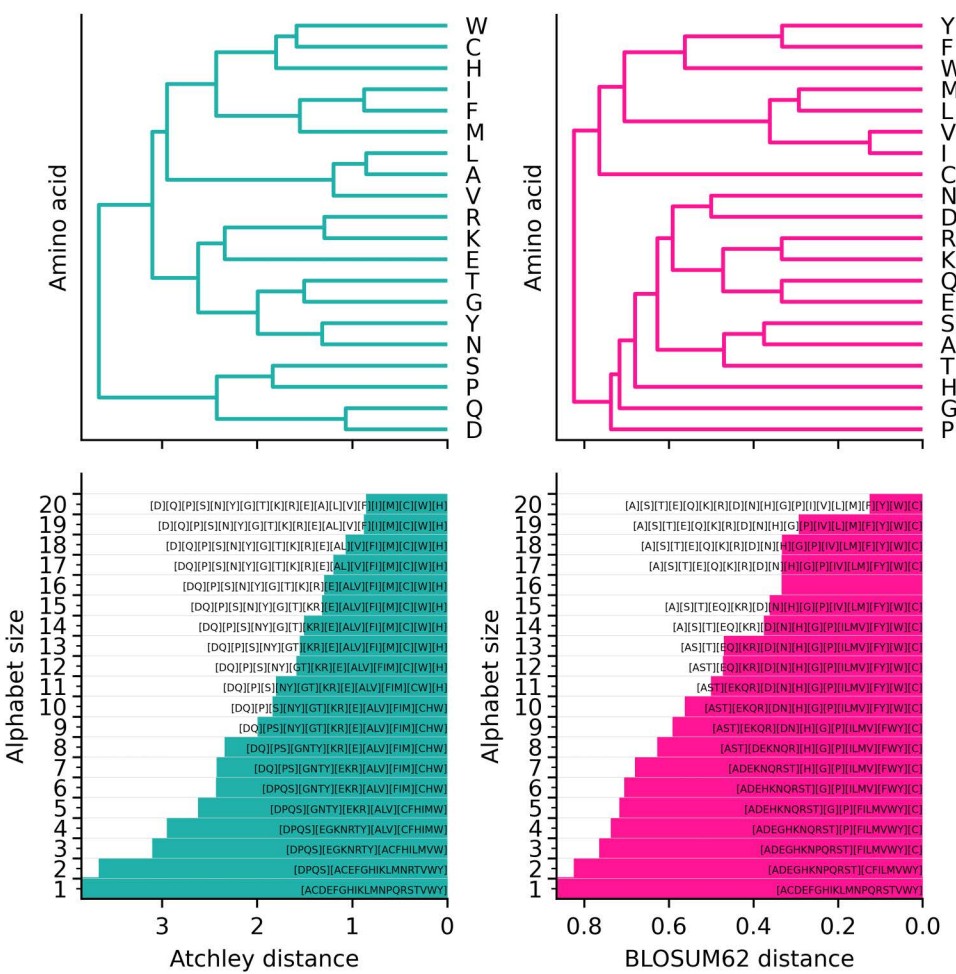

**Fig 1. Reduced alphabets of amino acids.** Atchley factor (left) and BLOSUM62-based (right) hierarchical clustering solutions for the 20 amino acids. Dendrogram top left and right show clusters of amino acids over a range of distance thresholds. Below, alphabet size is plotted over the distance threshold amino acid equivalence annotations. Amino acids considered equivalent within each alphabet are grouped using [].

## CMV classification problem

The TCR $\beta$ chain repertoire samples obtained from blood from 2 cohorts published by Emerson et al. were used as CMV training and testing datasets [4], which are summarised in Table 1. All 640 and 119 samples each within the training and testing datasets were downsampled to 23024 reads with no loss of samples as a result. Sample depth may differ by chance or in a systematic manner. If correlated with immune status, models may learn to use this information, although it is a product of technical variation such as degradation of genetic material or quantity of material sampled, and not reflective of biological differences. This downsampling is applied in order to avoid confounding by sample depth, wherein models may suffer from generalisability issues and may be misleading if interpreted. We do not expect this downsampling to greatly affect performance, as gradient boosted tree models show little deterioration in performance at a 0.1% sampling ratio for the same dataset [7]. In addition, kmer-based models and a deep learning method showed strong results on this dataset when downsampled to 10,000 CDR3 sequences [8].

XGBoost and logistic regression with an L1-norm or LASSO penalty are trained to classify $\beta$ chain TCR repertoires according to cytomegalovirus seropositivity, indicating past or current infection. These models are trained on 5 different

**Table 1. Summary of CMV datasets.**

|  | CMV training dataset | CMV testing dataset |
|---|---|---|
| Locus | TRB | TRB |
| Starting material | DNA from blood | DNA from blood |
| Samples before downsampling | 640 | 119 |
| CMV+ samples before downsampling | 289 | 51 |
| CMV- samples before downsampling | 351 | 68 |
| Mean reads before downsampling (min-max) | 1000893(23024 − 39775645) | 301631(44656 − 815687) |
| Downsampled reads to | 23024 | 23024 |
| Samples after downsampling | 640 | 119 |
| CMV+ samples after downsampling | 289 | 51 |
| CMV- samples after downsampling | 351 | 61 |

Mean reads before downsampling rounded to nearest integer.

kmer-based representations of the CMV training dataset and tested on previously unseen repertoires. The five representations, and up to 20 different alphabets within the reduced alphabet representations, may require different model hyperparameter settings such as regularisation strength in order to train an optimal model. To tailor models to each representation, we set XGBoost hyperparameters *reg_lambda*, *max_depth* and *learning_rate* and logistic regression hyperparameter *C*, using Bayesian optimisation.

6 performance measures are shown in Table 2 for XGBoost and logistic regression: area under receiver operating characteristic curve (AUROC), accuracy, sensitivity, specificity, balanced accuracy and Matthew's correlation coefficient (MCC). Accuracy, though shown for completeness, does not fairly represent performance due to class imbalance in the CMV datasets. Each of sensitivity and specificity alone does not fully capture model performance, but AUROC, balanced accuracy and MCC combine contributions from both of these measures into a single value. AUROC, shown in the first column of each table, is ultimately used to evaluate overall model performance for consistency with other work [7], and because it takes classification probability into account which is not the case for balanced accuracy or MCC. Additionally, model hyperparameters chosen through Bayesian optimisation are shown for each model in S1 Table.

**Table 2. Testing performance of XGBoost and logistic regression models on CMV testing dataset with five kmer representations.**

| Model | Features | Encoding | s | AUROC | Acc. | Sens. | Spec. | Bal. acc. | MCC |
|---|---|---|---|---|---|---|---|---|---|
| XGB | kmers |  |  | 0.748 | 0.706 | 0.647 | 0.750 | 0.699 | 0.398 |
| XGB | RA kmers | BLOSUM62 | 10 | **0.797** | 0.697 | 0.608 | **0.765** | 0.686 | 0.377 |
| XGB | RA kmers | Atchley | 9 | 0.790 | **0.714** | 0.667 | 0.750 | **0.708** | **0.417** |
| XGB | kmer clusters | BLOSUM62 |  | 0.712 | 0.647 | **0.765** | 0.559 | 0.662 | 0.324 |
| XGB | kmer clusters | Atchley |  | 0.716 | 0.647 | 0.745 | 0.574 | 0.659 | 0.318 |
| L1LR | kmers |  |  | 0.633 | 0.580 | 0.471 | 0.662 | 0.566 | 0.134 |
| L1LR | RA kmers | BLOSUM62 | 11 | 0.687 | 0.630 | 0.529 | 0.706 | 0.618 | 0.238 |
| L1LR | RA kmers | Atchley | 13 | 0.601 | 0.639 | 0.510 | 0.735 | 0.623 | 0.251 |
| L1LR | kmer clusters | BLOSUM62 |  | 0.681 | 0.538 | 0.765 | 0.368 | 0.566 | 0.142 |
| L1LR | kmer clusters | Atchley |  | 0.529 | 0.479 | 0.569 | 0.412 | 0.490 | −0.020 |

Each model is trained on entire CMV training dataset and tested on entire CMV testing dataset. Performance measures including AUROC, accuracy, sensitivity, specificity, balanced accuracy and MCC are reported for testing dataset.

PLOS Computational Biology

When all XGBoost models are trained on the CMV training dataset and tested on the CMV testing dataset, a BLOSUM62 reduced alphabet is estimated to give the highest AUROC of 0.797, but is not universally best. The model trained on Atchley reduced alphabet kmers also performs well in testing, resulting in the highest balanced accuracy of 0.708 and highest MCC of 0.417. Both reduced alphabet approaches outperform kmer cluster approaches in AUROC, accuracy, balanced accuracy and MCC. Logistic regression models perform poorly to in comparison to XGBoost, with BLOSUM62 reduced alphabet kmers resulting in highest AUROC. However, it is unclear whether any method outperforms the rest for logistic regression. Reduced alphabets were close in size of 9 and 10 for XGBoost models and 11 and 13 for logistic regression, shown in Table 2 and Fig 2, as chosen by best AUROC in cross-validation within the training of the model on the entire CMV training dataset. This halving of the amino acid alphabet appears not to be the only size that can lead to a strong XGBoost model; AUROC is consistenty high over a continuous range of alphabet sizes in Fig 2. Differing regularising hyperparameters are chosen for the five feature types as in S1 Table, and in particular *reg_lambda* and *C*, which are equivalent but inverse of one-another, imply stronger regularisation is applied to all features than the default setting of 1 for each.

**CMV XGBoost model runtime comparison**

We also evaluated computational efficiency of the kmer representations by calculating the training time for XGBoost models trained for the CMV classification problem. With reduced alphabet kmers and kmer clusters, the optimal amino acid similarity threshold was set through assessment of a set of hyperparameters designed to be roughly equivalent. In kmer clustering, the number of clusters is also the number of kmer groups, which is related to the amino acid similarity threshold at which we consider amino acids equivalent. With a reduced alphabet, the number of kmer groups arising from a reduced alphabet can be calculated as $s^k$, where $s$ is the size of the reduced alphabet. We try 18 similarity thresholds for each feature type, excluding the trivial cases where all kmers are in the same group ($s = 1$), as well as where each kmer is in its own group ($s = 20$). Since BLOSUM62 does not result in a reduced alphabet size of 16, only 17 thresholds could be tried for the BLOSUM62-encoded features. We used a set of fixed XGBoost hyperparameters for the purpose of recording runtime.

For all feature types, the total training time was recorded which shown in Fig 3 on both a logarithmic and its original scale. The kmer features result in by far the shortest training time. Reduced alphabet features result in over 1 order of magnitude increase in training time, but kmer clusters result in nearly 3 orders of magnitude

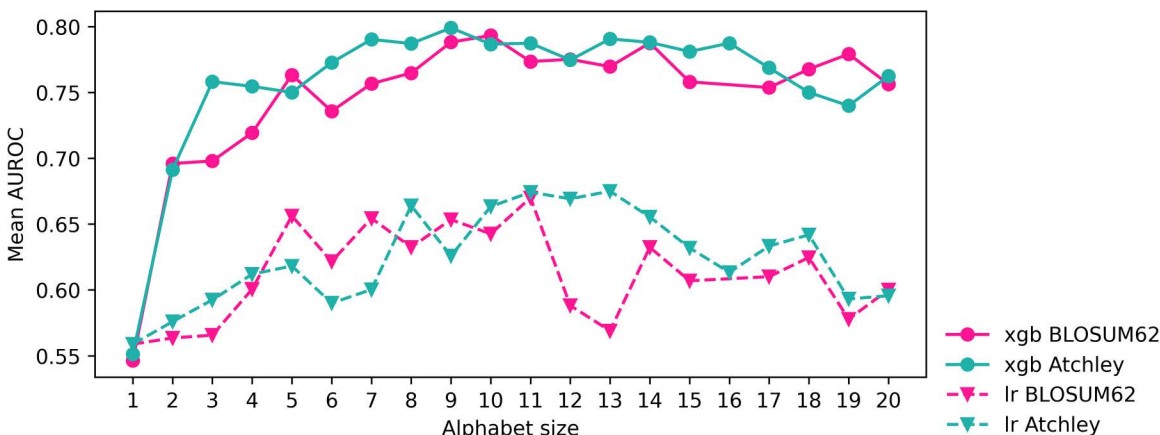

**Fig 2. AUROC of reduced alphabet sizes when evaluated in cross validation for XGBoost and logistic regression models on CMV training dataset.** Alphabet sizes chosen for final model are those with highest AUROC.

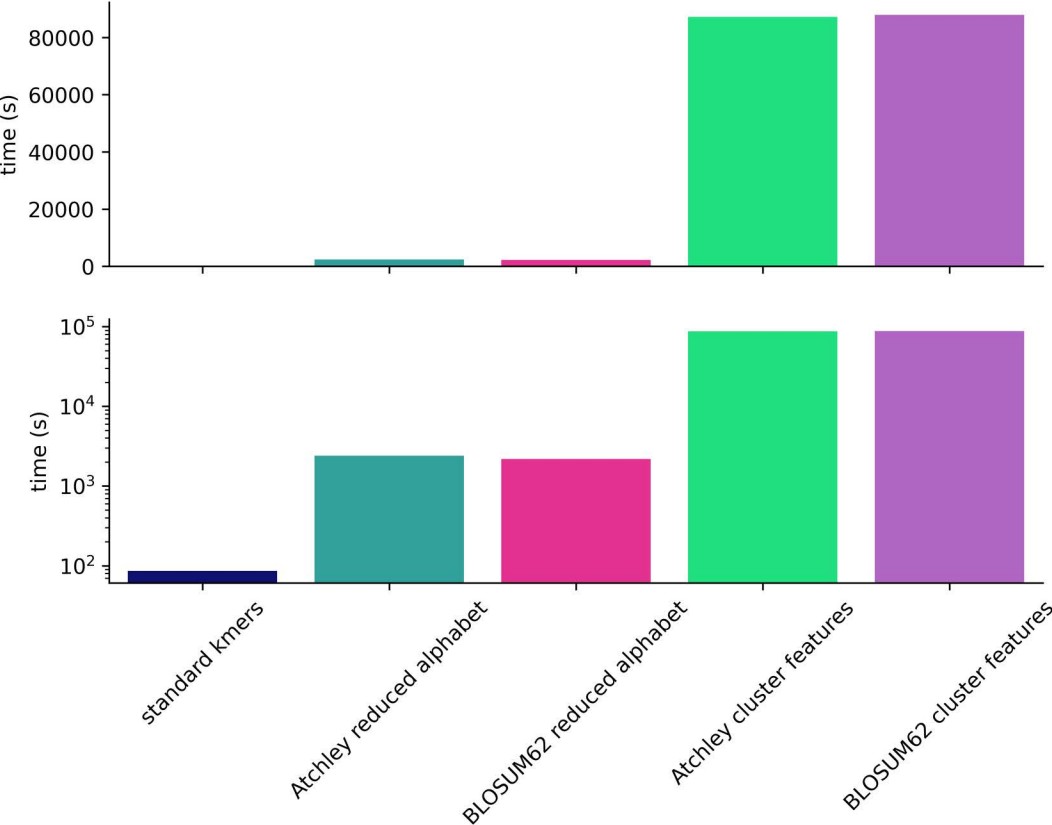

**Fig 3. Full training duration for 5 feature types with XGBoost model for CMV classification problem.**

increase over kmers. The greatest training time of 87888 seconds is equivalent to 1 day, 0 hours, 24 minutes and 48 seconds.

For reduced alphabet kmer and kmer cluster features, duration of each amino acid similarity hyperparameter evaluation in 5-fold cross validation within the model training process is also shown in Fig 4. Both reduced alphabet size and number of clusters have a non-linear relationship with time for each hyperparameter evaluation within the training process, which may be explained by the cross-validation involved. K-means clustering, which is expected to scale in proportion to the number of clusters [51], likely explains the much greater time for each evaluation for kmer cluster features over reduced alphabet features, where clustering is performed only once on the 20 amino acids. Increasing the kmer length would potentially increase the number of clusters for k-means if a wide range of amino acid similarities were to be explored, increasing training time potentially much beyond our result here for 4mers.

### Interpretation of CMV classification models

First, XGBoost and logistic regression CMV classification models are compared across feature types. Feature importance derived from each model is assigned to each kmer contained within a given feature so that each kmer member of a cluster or motif have the same importance as the cluster or motif feature. We utilise the built-in feature importance method within XGBoost, specifically the gain. For logistic regression, we take the absolute value of the feature coefficients. Correlation of the feature importance assigned to kmers is shown in Fig 5 for all CMV models. Aside from the logistic regression models with standard kmer and reduced alphabet kmer features, which we expect due to large alphabet sizes chosen

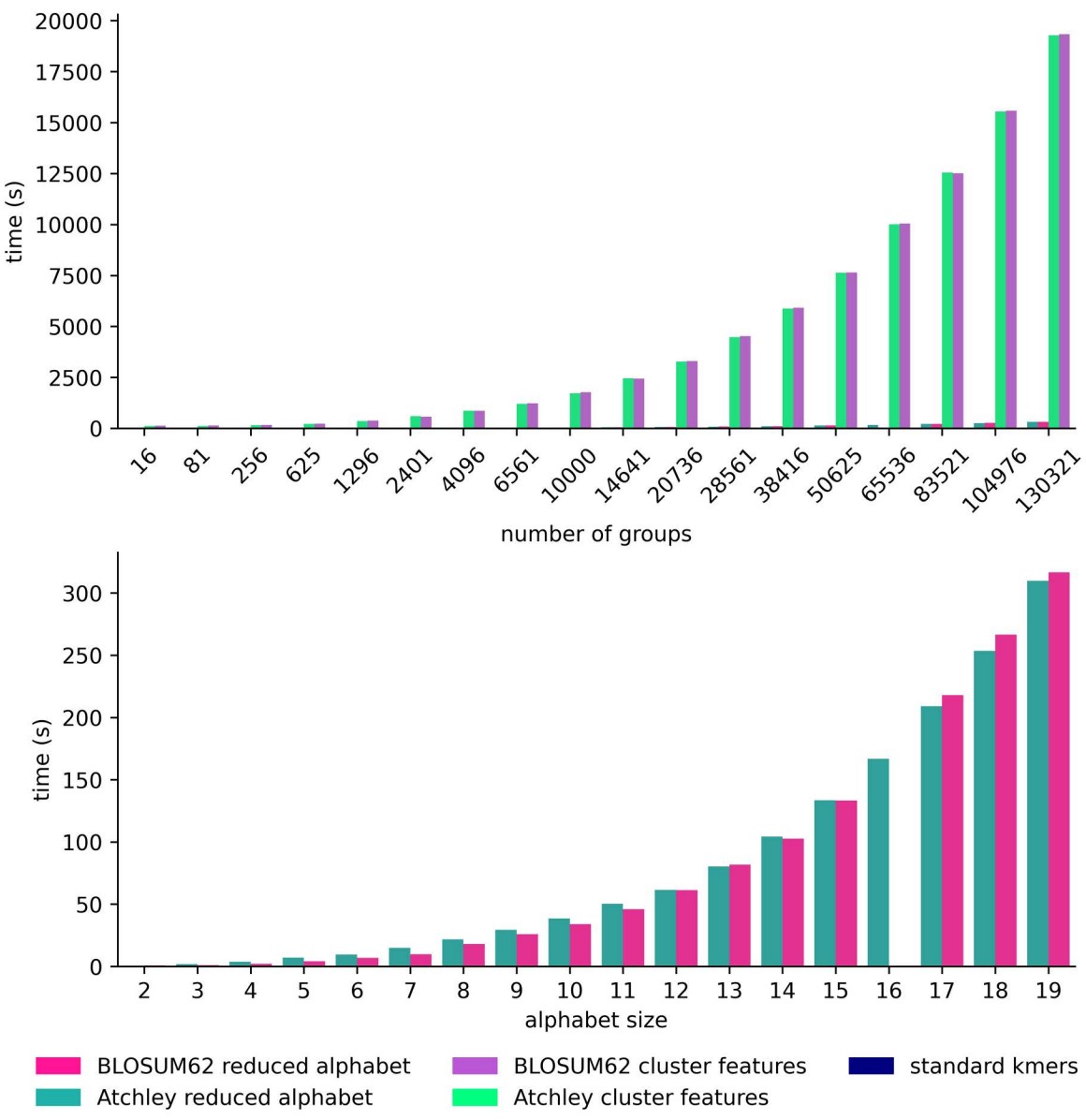

**Fig 4. Duration of each amino acid similarity-related hyperparameter evaluation in training of XGBoost model for CMV classification problem.**
Top: time for evaluation of each hyperparameter for the number of kmer groups defined by a hyperparameter. For cluster features this is number of clusters of kmers. For a reduced alphabet this is the number of motifs that could be observed when the alphabet is applied. Bottom: the same as Top for reduced alphabet features only with smaller vertical scale.

in training seen in Table 2, minimal positive correlation is observed. This implies that models have divergent classification bases.

For the best-performing models, we analyse the important features further. Fig 6 shows a summary of features including total number used, proportion of importance, and number of kmers included within each feature for the XGBoost models trained on the entire CMV training set. For standard kmer and reduced alphabet kmer features, 500–1500 or more features are incorporated into the model. Since the XGBoost model is regularised and performs better than a regularised

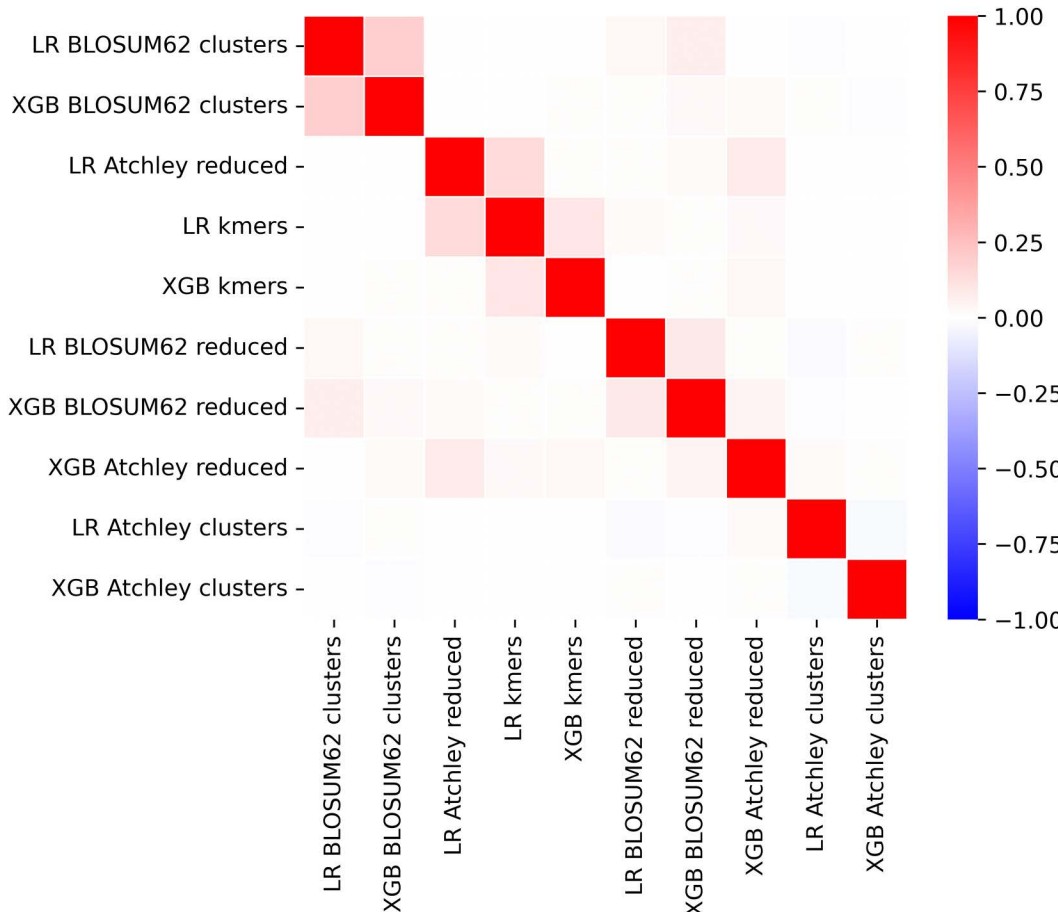

**Fig 5. Correlation of feature importance in XGBoost and logistic regression models for CMV status across 5 kmer-based representations.**

linear model, we expect that a complex relationship between the many kmer-based features is favoured for strong performance with standard kmers and reduced alphabet kmer representations, which seems likely given that few features have a share of more than 1% of the total feature importance. Kmer cluster features tend to contain large numbers of kmers, and fewer are utilised within a model. Since XGBoost models trained on these features perform nearly as well as the other features, it may be that there are multiple differing non-linear relationships that can capture some differences between TCR repertoires from individuals of each CMV status. A more detailed visualisation of features is included in S1 Fig, where kmers and kmer motifs that define each feature are annotated where possible.

Next, we sought to understand whether biological insights could be gained from interpretation of the importance assigned to kmer-based features. With 26658 CMV-specific CDR3 sequences from VDJdb [53] and 26658 CDR3 sequences not known to be CMV-specific sampled from the CMV-negative samples within the CMV testing dataset, the frequency of kmers and motifs are investigated. For the model with highest testing AUROC, XGBoost trained on BLOSUM62 reduced alphabet kmers, we used Fisher's exact test to quantify the differential presence of motifs between CMV-positive and negative TCR sequences used within the model. In Fig 7 many motifs are significantly differentially observed at a threshold of 0.05, indicated with stars. However, these motifs are not clearly the most important within the XGBoost model, and their normalised frequency differences, $\frac{f_{motif}^{CMV+} - f_{motif}^{CMV-}}{f_{motif}^{CMV+} + f_{motif}^{CMV-}}$) are relatively small and often negative. This lack of a clearly differentially present motif is consistent with poorer performance for the logistic regression model which is unable

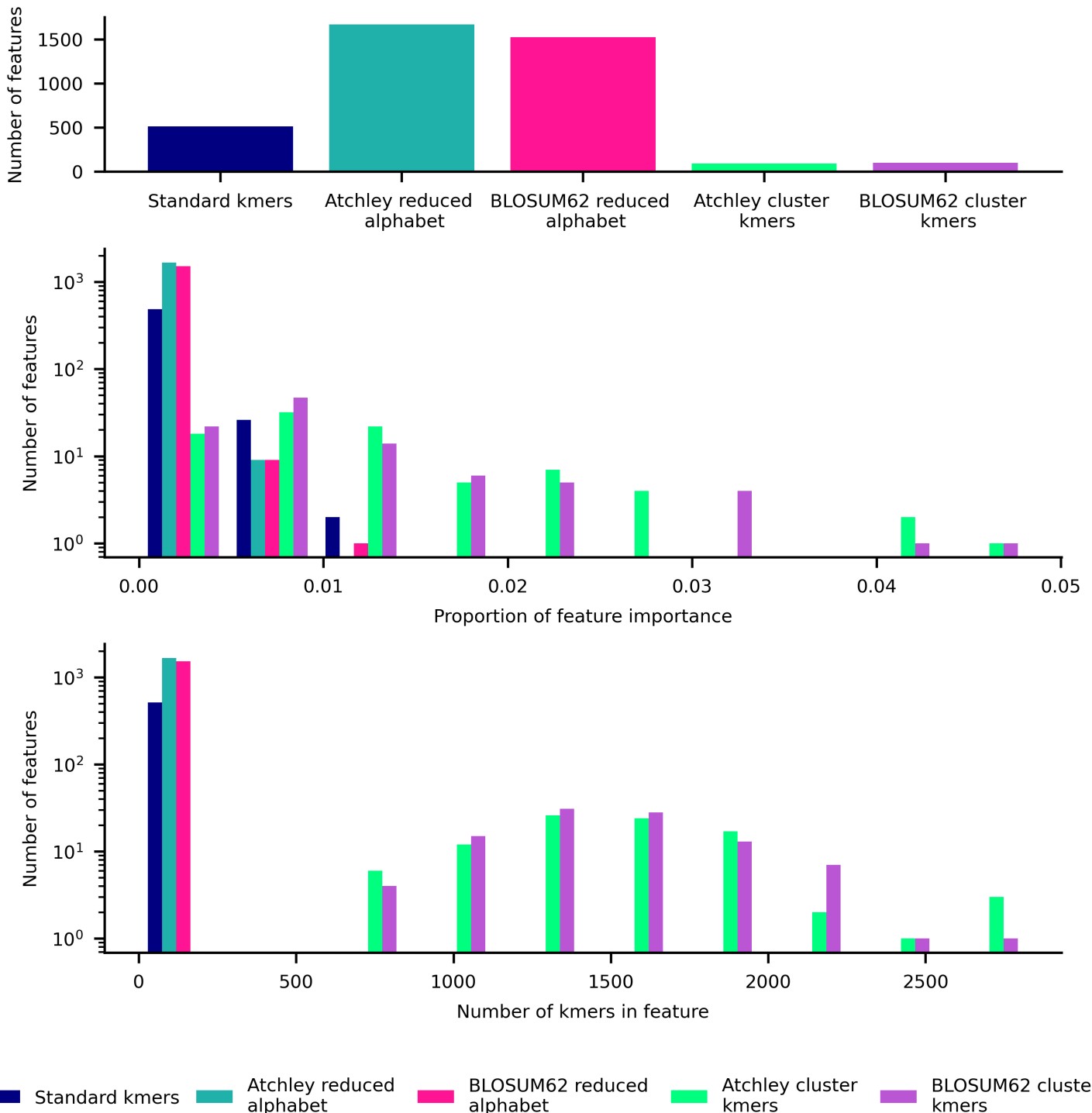

**Fig 6. Summary of features used in XGBoost model trained on CMV training dataset for each feature type.** Top: bar plot of number of features used in model. Middle: histogram of proportion of importance of features used in model. Lower: histogram of number of kmers within each feature used in model.

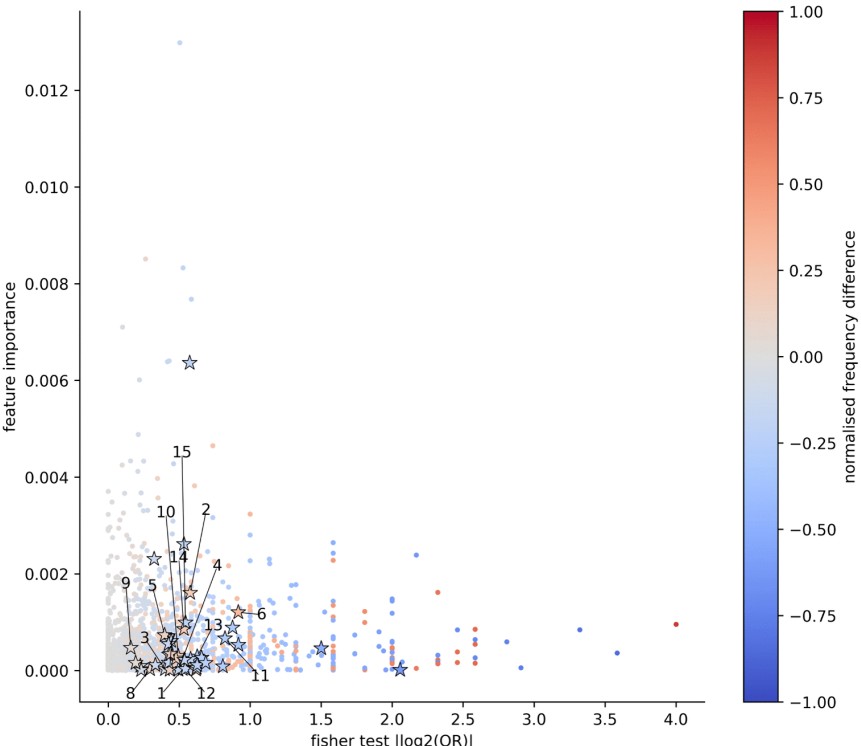

| | motif | OR | Padj |
|---|---|---|---|
| 1 | C[AST][AST][AST] | 1.44 | 1.12E-72 |
| 2 | [AST][AST][ILMV]G | 1.49 | 7.63E-23 |
| 3 | C[AST][AST][EKQR] | 0.758 | 9.17E-21 |
| 4 | [AST][AST][EKQR][EKQR] | 0.710 | 1.09E-18 |
| 5 | [DN][AST][EKQR][AST] | 1.32 | 2.04E-10 |
| 6 | G[FY][DN][EKQR] | 1.89 | 5.33E-09 |
| 7 | [AST][AST][ILMV][ILMV] | 1.35 | 8.03E-09 |
| 8 | [AST][AST][ILMV][AST] | 1.23 | 3.35E-06 |
| 9 | [EKQR][EKQR][FY][FY] | 1.12 | 3.94E-06 |
| 10 | [AST][AST]PG | 1.41 | 4.61E-06 |
| 11 | [EKQR][EKQR]G[EKQR] | 0.565 | 6.16E-06 |
| 12 | [AST][EKQR][EKQR][EKQR] | 0.686 | 5.78E-05 |
| 13 | [EKQR]G[EKQR][EKQR] | 0.647 | 1.76E-4 |
| 14 | [AST][ILMV]GG | 1.45 | 2.92E-4 |
| 15 | [EKQR][AST]G[EKQR] | 0.686 | 2.95E-4 |

**Fig 7. Scatter plot showing kmer feature importance, absolute value of log odds ratio from Fisher's exact test, and normalised difference in frequency for BLOSUM62 reduced alphabet motifs used in XGBoost model for CMV status.** Table shows top 15 full motifs with odds ratios and adjusted P-values; the full set of motifs is shown in S3 Table.

to characterise interactions between features. The XGBoost model trained on BLOSUM62 reduced alphabet kmers to distinguish TCR repertoires based on CMV status may perhaps capture some broad patterns of CDR3 amino acid properties of both TCRs that are able to interact with CMV epitopes, and those that do not.

## CeD classification problem

A CeD training dataset which includes 139 $\delta$ chain TCR repertoire samples was obtained from duodenal biopsies. Another published $\delta$ chain dataset from duodenal biopsies was used as a CeD testing dataset [28]. Some samples with low TCR read counts in the CeD training dataset are identified as a possible source of confounding. A downsampling threshold was therefore chosen by trading off the number of samples retained with the read counts of each downsampled sample, illustrated by Fig 8. The chosen downsampling threshold of 10078 was applied to both CeD datasets, resulting in a final count of 125 and 22 samples in each of the training and testing datasets respectively. The CeD training and testing datasets are summarised in Table 3.

XGBoost and logistic regression models trained on 5 different kmer-based representations of the CeD training dataset are evaluated in testing shown in Table 4. Model hyperparameters are set using Bayesian optimisation as for the CMV classification problem, resulting in final hyperparameters shown in S2 Table. Performance measures are considered in the same way as for the CMV classification problem, since class imbalance is also present in the CeD training dataset. TheBLOSUM62 reduced alphabet model results in the highest testing AUROC of 0.926, balanced accuracy of 0.864 and MCC of 0.730. The standard kmer model also results in strong testing performance with perfect specificity. Both kmer

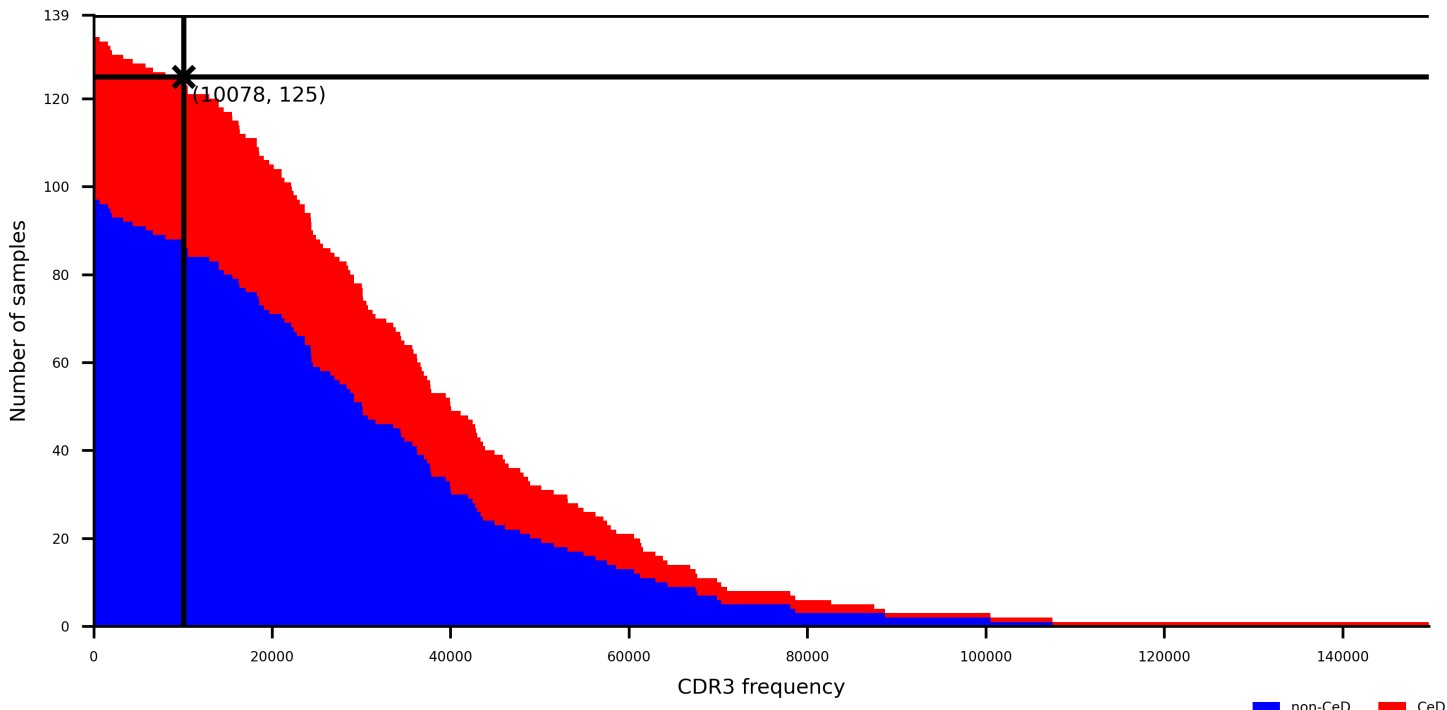

**Fig 8. Step plot showing number of CeD training dataset samples that have at least a CDR3 count shown on horizontal axis, distinguished by class.** A downsampling threshold is labelled with a cross at 10078 sequences, retaining 125 samples.

cluster features models result in lower AUROC than their matched reduced alphabet models, though for Atchley factors this difference is small. As shown in Table 4, the chosen BLOSUM62 reduced alphabet of size 14 is able to train a model that performs slightly better than or similarly to standard kmer features. For XGBoost models trained on reduced alphabet features, sizes from 6 to 15 may result in fairly consistent performance as shown in Fig 9. As for the CMV classification problem, different regularising hyperparameters are chosen for the five feature types shown in S2 Table. Again, *reg_lambda* and *C* imply stronger regularisation is applied to all features than the default, but perhaps to a lesser extent than for the CMV classification problem.

## Interpretation of CeD classification models

XGBoost and logistic regression CeD status classification models' feature importance are obtained as for the CMV classification models. In Fig 10 the 10 models are shown to be mostly uncorrelated in their kmer importance, apart from the kmer and reduced alphabet logistic regression models which is expected due to a large alphabet size of 18 in training on the whole CeD training dataset shown in Fig 9. For standard kmer and reduced alphabet kmer features, we see in Fig 11 that over 100 features are incorporated into each model which is roughly an order of magnitude lower than the CMV XGBoost model. Many standard and reduced alphabet kmer features have a greater share of the total feature importance than for the CMV classification model. Kmer cluster features have a more similar distribution to reduced alphabet features here, which may reflect the different amino acid similarity threshold captured in the clustering with 100 clusters but fewer kmers observed in the CeD training dataset than the CMV training dataset. Within the annotated stacked bar plot showing XGBoost feature importance as a proportion of the total in S2 Fig, we notice that SYWG which is the most important in the standard kmer model is included within the 5th most important feature in the Atchley reduced alphabet model,

**Table 3. Summary of CeD datasets.**

|  | CeD Training Dataset | CeD Testing Dataset |
|---|---|---|
| Locus | TRD | TRD |
| Starting material | DNA from duodenal biopsies | DNA from duodenal biopsies |
| Samples before downsampling | 139 | 23 |
| CeD samples before downsampling | 38 | 11 |
| Non-CeD samples before downsampling | 101 | 12 |
| Mean reads before downsampling (min-max) | 36049(39 − 149658) | 51979(7323 − 354606) |
| Downsampled reads to | 10078 | 10078 |
| Samples after downsampling | 125 | 22 |
| CeD samples after downsampling | 37 | 11 |
| Non-CeD samples after downsampling | 88 | 11 |

Mean reads before downsampling rounded to nearest integer.

**Table 4. Testing performance of XGBoost and logistic regression models on CeD testing dataset with five kmer representations.**

| Model | Features | Encoding | s | AUROC | Acc. | Sens. | Spec. | Bal. acc. | MCC |
|---|---|---|---|---|---|---|---|---|---|
| XGB | kmers |  |  | 0.909 | 0.818 | 0.636 | **1.00** | 0.818 | 0.683 |
| XGB | RA kmers | BLOSUM62 | 14 | **0.926** | **0.864** | 0.818 | 0.909 | **0.864** | **0.730** |
| XGB | RA kmers | Atchley | 12 | 0.810 | 0.727 | 0.727 | 0.727 | 0.727 | 0.455 |
| XGB | kmer clusters | BLOSUM62 |  | 0.769 | 0.727 | 0.636 | 0.818 | 0.727 | 0.462 |
| XGB | kmer clusters | Atchley |  | 0.793 | 0.818 | 0.818 | 0.818 | 0.818 | 0.636 |
| L1LR | kmers |  |  | 0.545 | 0.636 | 0.818 | 0.455 | 0.636 | 0.293 |
| L1LR | RA kmers | BLOSUM62 | 14 | 0.736 | 0.682 | 0.727 | 0.636 | 0.682 | 0.365 |
| L1LR | RA kmers | Atchley | 18 | 0.669 | 0.636 | 0.636 | 0.636 | 0.636 | 0.273 |
| L1LR | kmer clusters | BLOSUM62 |  | 0.719 | 0.727 | **0.909** | 0.545 | 0.727 | 0.488 |
| L1LR | kmer clusters | Atchley |  | 0.413 | 0.409 | 0.455 | 0.364 | 0.409 | −0.183 |

Each model is trained on entire CeD training dataset and tested on entire CeD testing dataset. Performance measures including AUROC, accuracy, sensitivity, specificity, balanced accuracy and MCC are reported for testing dataset.

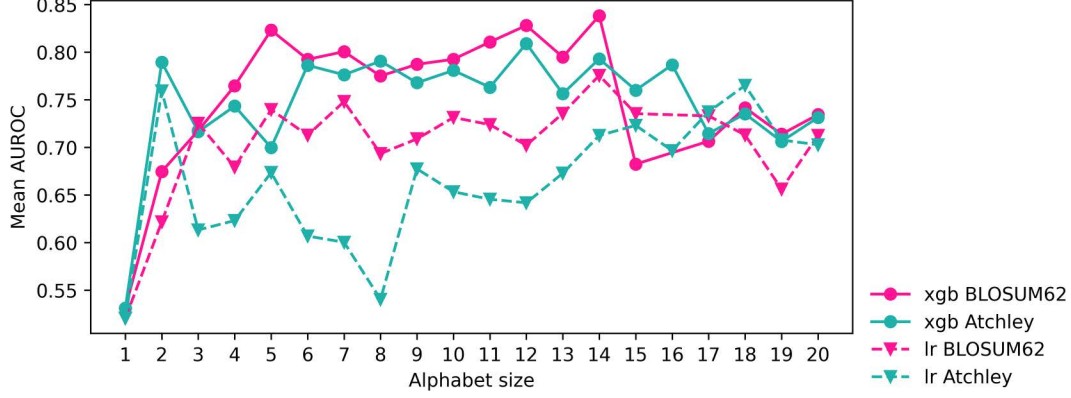

**Fig 9. AUROC of reduced alphabet sizes when evaluated in cross validation for XGBoost and logistic regression models on CeD training dataset.** Alphabet sizes chosen for final model are those with highest AUROC.

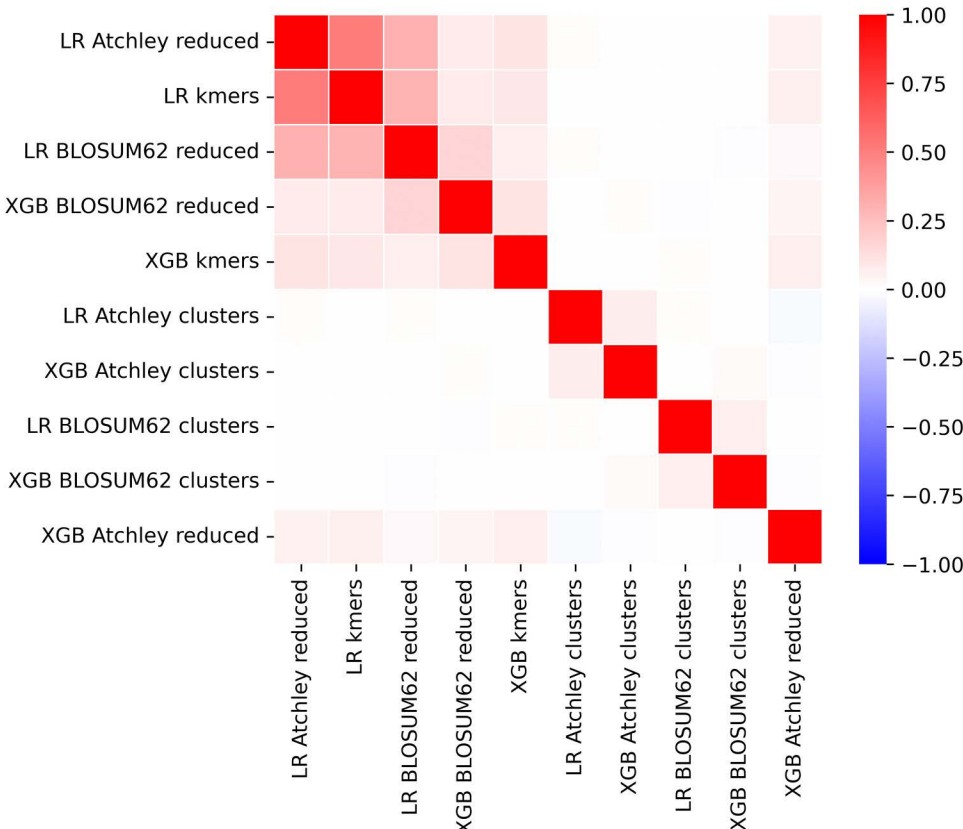

**Fig 10. Correlation of feature importance for XGBoost and logistic regression models trained on CeD training dataset across 5 kmer-based representations.**

S[NY][CW][GT], and the most important feature in the BLOSUM62 reduced alphabet model, S[FY]WG. However, largely the most important motifs within the reduced alphabet kmer models do not have obvious matches to the most important kmers within the standard kmer model, which is consistent with their correlation.

We observe that YWGI is ranked 20th in the standard kmer XGBoost model and overlapping with S[FY]WG which is the most important motif in the BLOSUM62 reduced alphabet model that performed best in testing. YWGI was previously reported by Han et al. in coeliac disease-associated TRD sequences [54]. We next conduct analysis to systematically investigate the important kmers within the model. In a similar manner to the CMV classification model investigation, we examine their presence in 186 TRD sequences from Han et al. [29,54] in contrast with 10806 TRD sequences we believe are not coeliac disease associated, obtained from non-coeliac samples from the CeD testing dataset not within the CeD associated TCR sequences from [54]. 3 significantly differentially present kmers are found according to Fisher's exact test at threshold of 0.05 and shown in Fig 12. PS[FY]W, S[FY]WG and CA[ILMV]G are relatively important in the model, have a non-negligible odds ratio and are more frequent in CeD-associated CDR3 sequences according to the normalised difference $\frac{f^{CeD+}_{motif} - f^{CeD-}_{motif}}{f^{CeD+}_{motif} + f^{CeD-}_{motif}}$). PS[FY]W and S[FY]WG are overlapping by 3 amino acids, and additionally captures 3 amino acids of YWGI. YWGI was reported in [54], therefore we may have reproduced the finding of a coeliac disease-associated motif, either extending or refining it.

To test the idea that the relationship between kmers within the XGBoost model may essentially stitch together kmers into a longer motif, we use a Mann-Whitney U test to assess whether the distributions of number of CeD-associated motifs

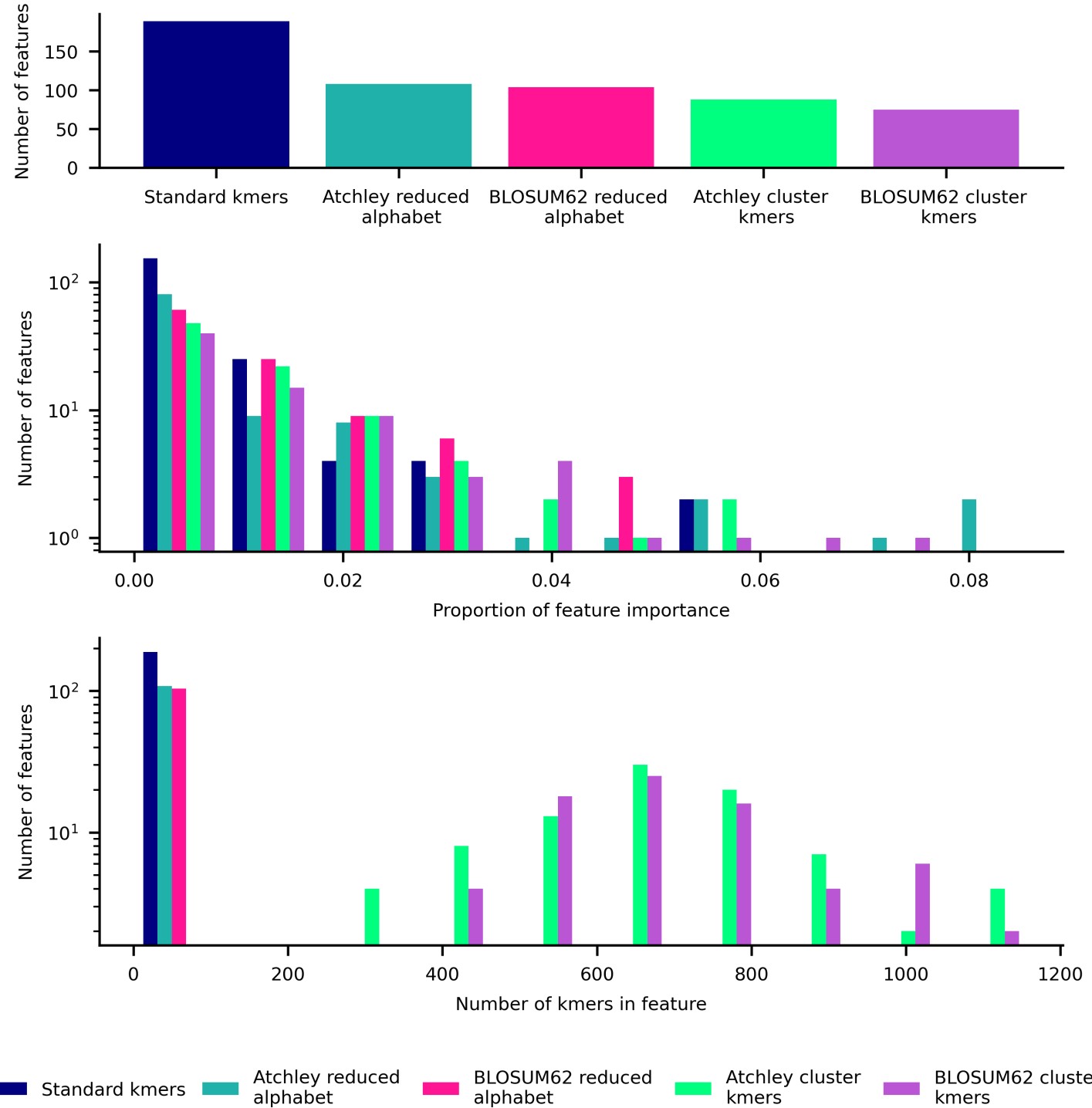

**Fig 11. Summary of features used in XGBoost model trained on CeD training dataset for each feature type.** Top: bar plot of number of features used in model. Middle: histogram of proportion of importance of features used in model. Lower: histogram of number of kmers within each feature used in model.

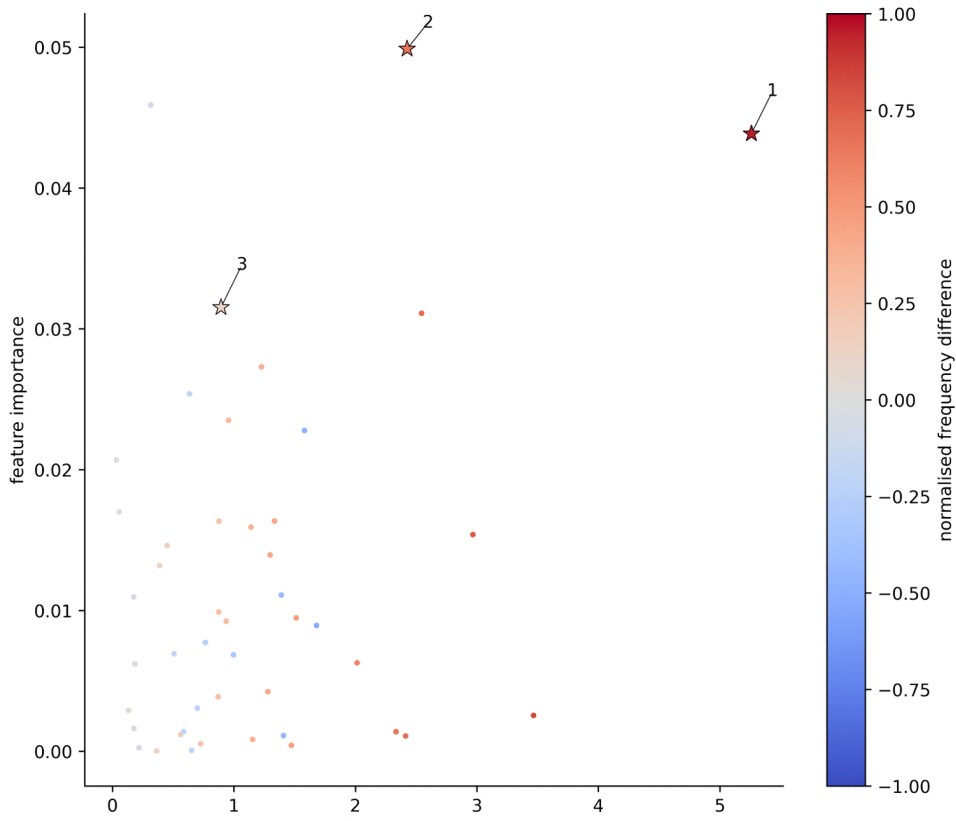

**Fig 12. Scatter plot showing kmer feature importance, absolute value of log odds ratio from fisher's exact test, and normalised difference in frequency for standard kmers used in XGBoost model for CeD status.** Table shows full motifs with odds ratios and adjusted P-values.

per CDR3 length differ between the two groups of CDR3 sequences, as in, significantly differentially present motifs with an odds ratio of greater than 1. S3 Fig shows a difference in medians between the distributions; distributions differ significantly with $P = 2.03^{-5}$, $U = 1182344.5$. This suggests that CeD-associated TRD CDR3 sequences are more likely to have higher proportions of their length covered by CeD-associated kmers. Also shown in S3 Fig are the CDR3 sequences towards the top of each distribution. At the top of the CeD-associated distribution is CALGERRPSYWGIRRGPLIF containing all 3 motifs. However, non-CeD-associated CDR3 sequences CALGELPSYWGLYTDKLIF, CALGELPSYWGYL-TAQLFF, CAPSYWGADKLIF, CALGVPSYWGIRGYTDKLIF and CALGELTPSYWGIRVSGLIF are all ranked higher or equivalently. It is important to contextualise their frequency: there are roughly 58 times as many non-CeD-associated sequences than CeD-associated sequences within this data, which can be seen in tails of the histograms in S3 Fig. In the high-ranked CeD-associated CDR3 sequences shown we often see PSYWGI or PSYWG covered by PS[FY]W, S[FY]WG and sometimes YWGI located near the centre of the sequence, which may suggest importance in binding.

## Discussion

We introduce a novel and flexible kmer representation of the TCR repertoire based on a reduced alphabet of amino acids. To understand the utility of adding amino acid similarity information to a kmer representation of the TCR repertoire for classification, our approach is evaluated in comparison to the kmer cluster representation by Thomas et al. and standard kmers. Atchley factor and BLOSUM62 amino acid similarity specifications are assessed for both reduced alphabet and

kmer clustering methods, and kmers of length 4 are used for all representations. To ensure that performance for each feature type is not due to suitability of model hyperparameter choice, we set model hyperparameters using Bayesian optimisation for each model and representation. The resulting 5 representations are evaluated in combination with XGBoost and logistic regression models for two different immune status classification problems: CMV infection and CeD status, which are each a means to provide evidence relating to immune status classification in general, as well as autoimmune disease status classification.

For the CMV datasets, standard 4mer XGBoost results very similar to those reported in the literature for 3mers are produced [7]. XGBoost performance exceeds that of logistic regression in testing for all representations, which may reflect the importance of interactions between frequencies of different 4mers. BLOSUM62 reduced alphabet kmers result in a CMV status XGBoost model that perform slightly better than standard kmers based on AUROC in testing. However, the highest AUROC of 0.797 is lower compared to 0.94 which is reported by the original authors for their method based on frequency of public CDR3 sequences [4]. This result by Emerson et al. is highly dependent on sample size beyond 250 [7]. We focus on kmer-based representation methodology for the purpose of application datasets smaller than 250 samples. While the potential of kmer based methods may be lower when the number of samples is high, they may enable unique insights to be gained in the data-poor setting through robustness to overfitting.

CeD XGBoost models generally perform well as might be expected given that CeD datasets originate from intestinal tissue for which TCR repertoire differences between CeD status have previously been reported [41]. Again, regularised logistic regression models for CeD do not reach the same AUROC, and have below the null value in some cases indicating poor discriminative ability. This implies that interactions between 4mers may be essential to train a TCR repertoire model to predict CeD status. Testing indicates that a BLOSUM62 reduced alphabet XGBoost model generalises better to the CeD testing set in terms of AUROC, although other models including the standard kmer model perform nearly as well in some or all performance measures. The CeD training dataset is imbalanced, the possible effects of which are mitigated by using performance measures not sensitive to class imbalance such as AUROC within cross validation for hyperparameter optimisation. The XGBoost models, which include class weighting to counteract class imbalance, were found to have high specificity and sensitivity, indicating that our strategies to handle the class imbalance were somewhat effective. However, we acknowledge that small size of the CeD testing dataset limits our confidence in the testing performance reported here.

Kmer cluster approaches do perform reasonably well across classification models when 100 clusters of 4mers are used to represent the TCR repertoire, which would be equivalent to an alphabet size of roughly 3. However, runtime experiments have shown that the reduced alphabet approach to applying amino acid similarity information is much more efficient, especially when evaluating high numbers of clusters, which is likely when exploring large kmer spaces in the case of 4mers or longer. The kmer clusters presented here are able to train models comparable to those by other kmer representations in some cases. However, the ability to efficiently set amino acid similarity using our novel reduced alphabet approach has greater potential to enable insights into suitability of amino acid encodings for TCR repertoire models to predict immune status.

The observation that utilising amino acid alphabets of size 14 or smaller leads to just as good or a better representation for discriminating TCR repertoires by immune status than the 20 amino acids may be useful in understanding the utility of kmer representations in general. It may be that since a reduced alphabet reduces the dimensionality of the kmer representation, its effect is that of regularisation which would lead to improve generalisability. Alternatively, small reduced alphabets may allow flexibility in certain kmer positions, since one of many amino acids could be present. This flexibility may resemble a gap character, in which any amino acid could be present. The coeliac disease-associated R-motif observed in $\alpha\beta$ TCR sequences contains gap characters [55], suggesting that a reduced alphabet approach may be a suitable tool for discovering other autoimmune disease-associated motifs. However, such gapped motifs could also be captured in a model by interactions between standard kmers than are not contiguous in a CDR3 sequence.

For CMV status classification, representations utilising BLOSUM62 lead to higher AUROC than for Atchley factor representations. However, this is not always true for CeD status models, and overall one encoding is not clearly more suitable than the other for TCR repertoire representation. The CMV datasets, being larger in size, may provide a higher quality of evidence for the most appropriate amino acid similarity specification for TCR sequences. It could also be the case that the most appropriate amino acid similarity specification for $\beta$ chain TCR sequences differs from $\delta$ chain sequences, since, in contrast to $\alpha\beta$ T cells, $\gamma\delta$ T cells are known to bind non-peptide antigens [56]. Further, properties or probable substitutions of amino acids that underlie function of TCRs may differ from those for proteins in general [57]. Any amino acid relationships that are discovered as important in TCR binding with peptide-MHC complexes or antigens in general would be computationally feasible to explore thoroughly in combination with a TCR repertoire classification model using our flexible reduced alphabet approach [47,57].

The reduced alphabets of amino acids give rise to kmer-based features that can be described with a motif with known underlying amino acid relationships. In comparison to kmer clusters, where clusters are defined by a cluster centre which is a vector of attributes for k amino acids in the kmer as well as its cluster boundary, we reason that our reduced alphabet approach is more readily interpretable. Reduced alphabets lead to different classification bases than standard kmers, as in, divergent kmers or motifs are utilised to make predictions, which could lead to alternative biological interpretations. However, the feature importance results obtained from XGBoost should be viewed with caution, since 5 different measures of importance: gain, total gain, cover, total cover and weight, could each be used which are inconsistent with each other [58]. Nonetheless, we were able to identify BLOSUM62-based motifs with a large share of feature importance in the CeD XGBoost model that also are significantly differentially present in CeD-associated TCR sequences from the literature. Two of these motifs were overlapping and more likely to be present in the CeD-associated sequences. One of the motifs ovelaps by 3 amino acids with a kmer that was previously proposed by the original authors within a CeD-associated motif [54]. We also identified differentially present BLOSUM62-defined motifs in CMV-specific TCR sequences which were utilised in the CMV XGBoost model, but believe that dedicated explanatory methods are required in order to understand the complex interactions between features, since many of these motifs are slightly less likely to be present in CMV-specific sequences. Explanatory methods would also be required to interpret deep learning methods for TCR repertoire classification, since they do not have inherent feature importance measures.

The distribution of CDR3 coverage by CeD-associated $\delta$ chain TCR motifs used by the XGBoost model differs in CeD-associated sequences opposed to non-CeD-associated sequences, with a higher median in the former. This suggests that, in the XGBoost model, there may be some relationship between kmer features that essentially assembles longer, possibly gapped, motifs. $\delta$ TCR chains from $\gamma\delta$ T cells have longer CDR3 regions than those in TCR $\beta$ chains of $\alpha\beta$ T cells [59], so inclusion of interaction terms that combine 4mers or 4mer motifs may allow greater capacity to characterise TCR specificity within a classification model. In the future, kmer length could be fit as a hyperparameter alongside model hyperparameters and alphabet size, which may improve performance and generalisability, and might lead to a model that is easier to interpret. Setting kmer length, as well as expanding the model hyperparameters explored within Bayesian optimisation, would be more computationally feasible with the reduced alphabet method over kmer clustering.

Overall, our work reveals important considerations when utilising an amino acid-aware kmer-based representation of the TCR repertoire in an immune status classification model. These results suggest some value of exploring TCR repertoire representations in a systematic manner, and provide a fast and flexible method for this exploration. We suggest that a modest benefit may be gained in certain cases when employing a reduced alphabet representation over both a standard kmer representation and one based on kmer clusters. In addition, results indicate that greatly reduced alphabets of amino acids can provide as useful of a classification basis as the full amino acid alphabet. The ability to set amino acid similarity as a hyperparameter with up to only 20 discrete values enables amino acid relationships within kmers to be explored for different properties and substitution matrices in a way that would be computationally challenging or impractical with a kmer

clustering approach. However, the utility of TCR repertoire classification methods as a means to gain understanding of disease will ultimately only be realised if interpretability and explainability are priotitised in future work.

## Materials and methods

### TCR repertoire datasets

4 TCR repertoire datasets are used to evaluate models in this work, including training and testing datasets for each of CMV infection status and CeD status classification problems.

### CMV datasets

Two cohorts of TCR repertoire samples published in [4] were downloaded from the ImmuneACCESS database. For both cohorts, TCR $\beta$ repertoires were obtained from genomic DNA in peripheral blood. Multiplex PCR was used for amplification, and sequencing was achieved using Illumina HiSeq. TCR repertoire samples were preprocessed by the original authors.

Cohort 1 was used as the CMV training dataset, and is stated to include 666 healthy bone marrow donors [4]. 665 samples were available for download, and 640 of these samples had CMV status labels. Two different column headings were identified as candidates from which to extract counts of each unique TCR sequence read, named *seq_reads* and *templates*. For 465 samples, only the *templates* column was present. For 123 samples, both *seq_reads* and *templates* headings were present, and both were populated with data. For another 76 samples, both headings existed but *templates* was not populated. A final single sample had both headings but missing values in the *templates* column. A decision to follow a procedure consistent with that by Katayama and Kobayashi [7], taking the *seq_reads* column if it exists and the *templates* column otherwise, was made for purposes of comparison.

Cohort 2 was used as the CeD testing dataset and includes healthy volunteers recruited to study infection [4]. 119 samples were available for download, which all included a *templates* columns rather than a *seq_reads* column.

### CeD datasets

The CeD training dataset includes TCR $\delta$ chain sequences from fully anonymised duodenal samples diagnosed with either active coeliac disease or diagnosed as normal (Ethics Ref: 04/Q1604/21, IRAS reference: 162057). DNA was extracted from the formalin-fixed paraffin embedded (FFPE) biopsy samples and bulk TCR $\delta$ chain libraries were created using the TCRD Gene clonality master mix (Invivoscribe) and then the Truseq Kit (Illumina). The final pooled library was sequenced by Illumina Miseq with 300 cycles. Paired end reads were preprocessed using the MiXCR align command [60].

Another $\delta$ chain dataset with coeliac disease status labels was previously obtained using similar methodology [28] and used as a CeD testing dataset. TCR $\delta$ chain sequences were obtained with DNA from FFPE duodenal biopsies. Despite some differences in preprocessing operations compared with the CeD training dataset, it is believed that this dataset can serve as a testing set due to similarity of preparation and sequencing.

In support of this publication, both CeD training and testing datasets are available at ImmPort (https://www.immport.org) under Study Accession SDY2976.

### TCR datasets of known immune status association

Additionally, we interpret final, trained models by comparing frequency of features in TCRs associated with and, to the best of our knowledge, not associated with the immune status.

### CMV-specific TCR sequences

Human $\beta$ chain TCR CDR3 sequences with epitope species CMV were downloaded from VDJdb [53] on 11th October 2025. Duplicate amino acid CDR3 sequences were removed, as were any not starting with C or ending with F or W.

We also removed CDR3 sequences with length less than 11 or more than 32. 26658 CMV-specific CDR3 sequences remained after filtering.

### Likely not CMV-specific TCR sequences

Since CMV testing data was not used to train the model in any way, we take CDR3 sequences from samples from donors who are seronegative for CMV. First we remove any CDR3 sequences present in the CMV-specific TCR sequences. As with the CMV-specific TCR sequences we also filter out duplicates, CDR3 sequences not starting with C or ending with F or W, and those with length less than 11 and more than 32. Since this resulted in 9,616,524 sequences, we randomly sampled without replacement to obtain 26,658 likely non-CMV-specific TCR sequences, an equal number to those CMV-specific.

### CeD-associated TCR sequences

Though there are many CeD-associated TCR sequences including those specific for gluten-derived peptides [29], these are mostly $\alpha$ or $\beta$ chains. However, in one study authors find a population of intestinal $\gamma\delta$T cells that expand after gluten challenge in coeliac disease patients, and sequence $\delta$ chain TCR repertoires [54], providing a source of known CeD-associated $\delta$ chain TCR sequences [29]. We take the CDR3 sequences and remove duplicates. We perform quality control as for CMV-associated sequences: remove those not starting with C or ending F or W; remove those with length less than 11 or more than 32. 186 CeD associated $\delta$ chain CDR3 sequences are obtained.

### Likely non-CeD-associated TCR sequences

In a similar approach to the likely non-specific TCR sequences, we use the non-coeliac donor samples from the CeD testing dataset. After removing duplicates, any within the CeD-associated sequences, those not starting with C or ending F or W and with length less than 11 or more than 32, we obtain 10806 CDR3 sequences.

### Representing the TCR repertoire with kmer features

TCR repertoire datasets are represented with 5 different kmer definitions. Unaltered kmers, referred to as standard kmers throughout and described below, are evaluated as a null representation by which to compare representations that incorporate amino acid similarity information. Two different specifications of amino acid information are added to this kmer representation using both a reduced alphabet and kmer clusters as described by Thomas et al. [46]. Code used to generate these TCR repertoire representations, and subsequently to perform classification, are available at https://github.com/hannrko/enc_kmer_tcr_models.

### Kmers

A kmer or k-mer describes a short section of a sequence with length k, where kmers of a sequence overlap. In this work, kmers that are used to represent the TCR are made up of 4 amino acids, and each 4mer overlaps the previous 4mer by all but one amino acid. Evidence suggests that for $\alpha\beta$ T cells, CDR3 positions in contact with antigenic peptides include 4 amino acids on average [43]. The choice of k as 4 throughout this work should therefore be appropriate for $\beta$ chain CMV datasets. For $\gamma\delta$ T cells, 4mers have been shown to separate samples in the CeD testing dataset by CeD status optimally [28], and so it is expected that 4mers should also be an appropriate representation for both CeD datasets.

### Amino acid encodings

Atchley et al. reduced a set of 54 recorded amino acid properties to 5 properties derived via factor analysis [48], which are termed Atchley factors. Substitution matrices such as BLOcks SUbstitution Matrices (BLOSUM) represent the chance that

occurrence of amino acids at the same position in a pair of sequences is the result of relatedness rather than randomness [52,61]. To represent amino acid similarities within kmers, both Atchley factors and BLOSUM62 are selected due to their frequent application to TCR repertoire representations [10,11,15,42–46,62,63].

## Kmers defined with a reduced alphabet

Reduced alphabets of amino acids have been used to represent proteins in classification problems [64], where groups of similar amino acids are considered equivalent and represented by the same character. In this work, kmers are defined using a reduced alphabet of amino acids with variable size. The reduced alphabet is based upon grouping of the 20 naturally-occurring amino acids which is achieved through hierarchical clustering. Starting from a distance matrix representing pairwise distances between all 20 amino acids hierarchical clustering solutions are calculated using an average linkage policy, using SciPy [65] (version 1.11.4) functions from scipy.cluster.hierarchy. Over a full range of distance thresholds, each possible clustering solution is recorded to produce multiple reduced alphabets. Amino acids in all kmers are replaced by a character that is representative of a cluster of amino acids. Frequencies of equivalent kmers that result are combined additively within each TCR repertoire sample.

The choice of reduced alphabet size can be treated as a hyperparameter. Since all reduced alphabets are stored in memory, only a single initial instance of hierarchical clustering is required to explore a full range of amino acid similarity thresholds. Aside from its size, the reduced alphabet representation is independent of the kmers that are observed in a TCR repertoire, since it is defined only on the basis of the 20 amino acids we expect to observe. This is in contrast to the kmer clustering described below, for which a clustering solution is dependent on kmers that are present in a sample, and that must be calculated separately for each similarity threshold specified.

Pairwise distances between amino acids are obtained in order to calculate the hierarchical clustering solutions used in this reduced alphabet approach. An Atchley factor-based distance matrix is defined through pairwise Euclidean distance of amino acids encoded by Atchley factors with standard scaling applied. A subset of BLOSUM62 corresponding to the 20 naturally-occurring amino acids was converted to a distance matrix. First, all elements of the matrix were made positive by subtracting the minimum value. Next, the matrix was normalised to have diagonal values of 1, as would be expected for a similarity matrix. To achieve this, each element representing substitution score of one amino acid for another was divided by the squared product of the self-substitution score of each amino acid. Finally, a distance matrix with diagonal elements of zero was retrieved by subtracting each element from 1.

## Kmer clusters

Kmer clustering is based on the approach described by Thomas et al. [46]. Unique kmers observed within a TCR repertoire training dataset are encoded by a vector of amino acid properties or features. The kmers are clustered into a specified number of clusters using K-means clustering with scikit-learn function *sklearn.cluster.KMeans* (version 1.4.2) [66] with all defaults except *n_clusters*. Kmers belonging to the same cluster are combined into the same feature additively, so that a TCR repertoire can be represented by the total counts of all kmers that map to each cluster. Kmers not observed in the training data are assigned to a cluster using the predict method. The number of clusters could conceivably be set as a hyperparameter, but the computational complexity of repeated K-means clustering limits the range of kmer similarity that can feasibly be explored. In this work, the number of clusters is set at 100 since strong performance was achieved by clustering 4mers into 100 groups by Thomas et al. [46].

Kmer clustering, in contrast to a reduced alphabet approach, takes encoded kmers as input. Atchley factors may be used to directly encode kmers as a vector for application of the kmeans clustering algorithm. Atchley factors are standardised prior to encoding, where each of 5 factors are transformed to have zero mean and unit standard deviation. The scaling applied should enable each factor to be considered in similar proportions during clustering. BLOSUM62 is converted to an encoding in a similar approach to that by Zhang et al. [42], where multidimensional scaling is applied to the

BLOSUM62-derived distance matrix. 5 features from the result of multidimensional scaling are utilised for consistency with the Atchley factor encoding. The values of each of the five features are used to encode each amino acid in a kmer as with Atchley factors.

## Classification methods

### XGBoost models

Katayama and Kobayashi demonstrate that an improvement can be gained by training non-linear model over a linear model on a TCR repertoires represented by kmers [7]. We also aim to understand the benefit of a non-linear model with inclusion of amino acid similarity information, and therefore, we use gradient-boosted trees similar to [7]. Gradient-boosted trees were implemented using XGBoost (version 2.0.3) classification method XGBClassifier [67]. For all XGBoost models, we set *scale_pos_weight* of negative to positive class ratio as recommended in the documentation to reduce the effects of class imbalance. For evaluation of classification performance we employed a hyperparameter optimisation strategy for *reg_lambda*, *max_depth*, and *learning_rate*. In the runtime experiment, we set *reg_lambda* = 1, *max_depth* = 10 and *learning_rate* = 0.1. Otherwise, all other defaults were used.

### Regularised logistic regression models

To understand the effect of non-linearity on the TCR repertoire classification in general, but also on the best amino acid similarity thresholds chosen for the different kmer features, we use regularised logistic regression. For regularisation we choose the L1 norm, also known as the LASSO penalty, which shrinks coefficients to zero which reduces the chance of overfitting. This is desirable, as a model with fewer used variables may be more interpretable. L1 norm-penalised logistic regression is implemented using scikit-learn function *sklearn.linear_model.LogisticRegression* (version 1.4.2). We used a hyperparameter optimisation strategy to set *C*, and otherwise set *penalty* = *l*1, *solver* = *liblinear*, *class_weight* = *balanced* and all other defaults.

### Model hyperparameter optimisation

Model hyperparameters are set using Optuna which implements Bayesian optimisation [69]. For each classification model, training data is split into a training fold with 80% of samples, and a validation fold of 20% of samples. A model is iteratively trained on the training fold with variable hyperparameters, and the validation fold is used to calculate AUROC as the objective for Bayesian optimisation to maximise. We performed 50 trials for each model. For XGBoost hyperparameters, ranges from which they are trialled are $1 - 100$ for *reg_lambda*, $3 - 10$ for *max_depth*, and $0.01 - 0.1$ for *learning_rate*. Logistic regression hyperparameter *C* is set within range $0.01 - 1.$, which is consistent with *reg_lambda* given that they are inversely equivalent. For all hyperparameters except *max_depth* which was set as an integer, we use *log* = *True* with float type.

### Optimisation of reduced alphabet size

For kmers with a reduced alphabet, the amino acid alphabet size was set based on performance on the training data. The performance of a model trained with each alphabet size, from 1 to 20, was evaluated using 5-fold cross validation with random shuffling. Performance is assessed using AUROC, where the alphabet size corresponding to the maximal value was chosen. If multiple optima were found, the smallest alphabet was selected from these, since there is a prior belief that reducing the dimensionality of the kmer representation will lead to less overfitting and improved performance on samples not in the training dataset. Additionally, model hyperparameters are optimised for each alphabet size, resulting in a nested validation strategy.

### Evaluation of TCR repertoire classification models

The five different kmer representations of the TCR repertoire are assessed using TCR repertoire datasets including a training and testing set with cytomegalovirus status labels, as well as a training and testing set with coeliac disease status

labels. TCR repertoire classification models are assessed using testing on separate testing datasets and 5-fold cross-validation repeated 3 times on training datasets. Performance measures, including AUROC, accuracy, sensitivity, specificity, balanced accuracy, and Matthew's correlation coefficient, are recorded for all test folds. When performance is reported for cross-validation, scores are combined across folds by calculating the mean and standard deviation. In each training and testing fold, steps were taken to ensure no leakage of data, as follows: standard scaling was applied to features of all data, with mean and variance calculated using only data in the training folds; the size of reduced alphabet used to define kmers is set only based on performance on the data in training folds; kmers used to define kmer clusters were limited to those observed at least once within the data in the training folds; and testing data was held out until model development was finalised. Plots used to visualise classification model results are generated using Matplotlib [68].

### Runtime experiment

For the CMV XGBoost models with 5 different feature types, we recorded full training time and, where applicable, the time to evaluate each amino acid similarity hyperparameter. The CMV classification problem was chosen for this experiment because of the CMV dataset's frequent reuse and previous runtimes reported. We aim to evaluate a fair range of amino acid similarity hyperparameters for the reduced alphabet and kmer cluster approaches, and made an assumption that dividing kmers into the same number of groups using either a reduced alphabet or clustering would result in a similar threshold of similarity at which we find amino acids to be equivalent. We therefore convert alphabet sizes or 2–19, excluding trivial cases 1 and 20, to number of groups with $n_g = s^k$ where $n_g$ is the number of groups and $s$ is the size of the reduced alphabet. These alphabet sizes converted to number of groups are used to set the possible values for number of clusters. XGBoost models are trained for CMV status classification with standard kmers, Atchley and BLOSUM62 reduced alphabet kmers where optimal alphabet size is chosen from values 2–19, as well as Atchley and BLOSUM62 kmer clusters where the optimal number of clusters is chosen from 18 discrete values from 16 to 130321 inclusive. XGBoost hyperparameters are otherwise chosen as previously described. This runtime experiment was undertaken with a CPU with 16GB memory with Intel Core i5-8500T processor.

### Interpretation of classification models

To systematically compare the presence of model-derived kmers of motifs of importance to a classification problem within TCR sequences of known immune status association, we implement Fisher's exact test. Within the model with best AUROC, each utilised feature is compared between TCR sequences known to be associated with the immune status, and those that are likely not, which will in following text be described as positive and negative TCRs respectively. We apply a Bonferroni correction to Fisher's exact test results, and consider significance at a threshold of 0.05. For the CeD model, we also apply a Mann-Whitney U test to compare the distribution of number of significantly differentially present kmers (in more CeD-associated CDR3 sequences than non-CeD-associated sequences) between positive and negative TCRs, normalised by CDR3 length.

### Supporting information

**S1 Table. Hyperparameters for XGBoost and logistic regression models trained on CMV testing dataset with five kmer representations.** Chosen hyperparameters for XGBoost models including *reg_lambda*, *max_depth* and *learning_rate* and for logistic regression models limited to the regularisation hyperparameter C trained on the CMV training dastaset.
(PDF)

**S2 Table. Hyperparameters for XGBoost and logistic regression models trained on CeD testing dataset with five kmer representations.** Chosen hyperparameters for XGBoost models including *reg_lambda*, *max_depth* and

*learning_rate* and for logistic regression models limited to the regularisation hyperparameter C trained on the CeD training dataset.
(PDF)

**S3 Table. Differentially present BLOSUM62-derived 4mer motifs in CMV-specific and likely non-CMV-specific CDR3 sequences.** Table shows full motifs with odds ratios and adjusted P-values.
(PDF)

**S1 Fig. Importance of features of XGBoost model trained on CMV training dataset for each feature type.** Feature importance, measured by gain, is shown in a stacked bar plot for each feature as a proportion of the total feature importance. Only features with at least 1% of total model importance are indicated by annotation. Background colour shows number of kmers included in each feature.
(TIF)

**S2 Fig. Importance of features of XGBoost model trained on CeD training dataset for each feature type.** Feature importance, measured by gain, is shown in a stacked bar plot for each feature as a proportion of the total feature importance. Only features with at least 1% of total model importance are indicated by annotation. Background colour shows number of kmers included in each feature.
(TIF)

**S3 Fig. Distributions of number of CeD-associated motifs per CDR3 length Histograms showing number of CeD-associated motifs per CDR3 length in CeD and non-CeD-associated CDR3 sequences.** 40 CDR3 sequences with highest number of CeD-associated motifs per length are listed in table.
(TIF)

## Acknowledgments

We would like to thank Yang Luo, Esther Ng, Jennifer Astley, Ruth Nanjala and Uzma Basit Khan for helpful discussions.

## Author contributions

**Conceptualization:** Hannah Kockelbergh, Liam Brierley, Peter L. Green, Andrea L. Jorgensen, Elizabeth J. Soilleux, Anna Fowler.

**Data curation:** Shelley C. Evans.

**Formal analysis:** Hannah Kockelbergh, Anna Fowler.

**Funding acquisition:** Elizabeth J. Soilleux, Anna Fowler.

**Investigation:** Hannah Kockelbergh.

**Methodology:** Hannah Kockelbergh, Liam Brierley, Anna Fowler.

**Resources:** Shelley C. Evans, Elizabeth J. Soilleux.

**Software:** Hannah Kockelbergh.

**Supervision:** Liam Brierley, Peter L. Green, Andrea L. Jorgensen, Elizabeth J. Soilleux, Anna Fowler.

**Visualization:** Hannah Kockelbergh.

**Writing – original draft:** Hannah Kockelbergh.

**Writing – review & editing:** Hannah Kockelbergh, Shelley C. Evans, Liam Brierley, Peter L. Green, Andrea L. Jorgensen, Elizabeth J. Soilleux, Anna Fowler.

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
