## [Decision Letter · Decision Letter 0]

11 Sep 2025

PCOMPBIOL-D-25-01274

Evaluating the utility of amino acid similarity-aware kmers to represent TCR repertoires for classification

PLOS Computational Biology

Dear Dr. Fowler,

Thank you for submitting your manuscript to PLOS Computational Biology. After careful consideration, we feel that it has merit but does not fully meet PLOS Computational Biology's publication criteria as it currently stands. Therefore, we invite you to submit a revised version of the manuscript that addresses the points raised during the review process.

Please submit your revised manuscript within 60 days Nov 11 2025 11:59PM. If you will need more time than this to complete your revisions, please reply to this message or contact the journal office at ploscompbiol@plos.org. Please include the following items when submitting your revised manuscript:

We look forward to receiving your revised manuscript.

Kind regards,

Shanfeng Zhu, Ph.D.

Academic Editor

PLOS Computational Biology

Mark Tanaka

Section Editor

PLOS Computational Biology

**Additional Editor Comments :**

In this study, the authors investigate the use of reduced amino acid alphabets for kmer-based classification of T-cell receptor repertoires. While the concept of applying biologically-informed clustering to simplify sequence representation is interesting and potentially valuable, the reviewers raise significant concerns that currently limit the paper's impact. The core issue is the lack of compelling evidence for the proposed method's advantage. The performance improvements over standard kmer baselines are marginal and not statistically significant (R1, R3), with one test case (CeD) showing worse performance(R2). In addition, the reviewers have substantial concerns over the robustness and generalizability, methodological rigor, computational efficiency, as well as biological interpretation. Please try to address these issues in the revisions.

**Journal Requirements:**

At this stage, the following Authors/Authors require contributions: Shelley C. Evans, Liam Brierley, Peter L. Green, Andrea L. Jorgensen, Elizabeth J. Soilleux, Hannah Kockelbergh, and Anna Fowler. Please ensure that the full contributions of each author are acknowledged in the "Add/Edit/Remove Authors" section of our submission form.

4) We notice that your supplementary Figures are included in the manuscript file. Please remove them and upload them with the file type 'Supporting Information'. Please ensure that each Supporting Information file has a legend listed in the manuscript after the references list.

5) In the online submission form, you indicated that "Cytomegalovirus datasets are made publicly available by the original authors." Please amend your Data Availability Statement to include the links to the datasets.

7) Please ensure that the funders and grant numbers match between the Financial Disclosure field and the Funding Information tab in your submission form. Note that the funders must be provided in the same order in both places as well. Currently, the order of the funders is different in both places.

8) Please provide a completed 'Competing Interests' statement, including any COIs declared by your co-authors. If you have no competing interests to declare, please state "The authors have declared that no competing interests exist". Otherwise please declare all competing interests beginning with the statement "I have read the journal's policy and the authors of this manuscript have the following competing interests:"

**Reviewers' comments:**

Reviewer's Responses to Questions

Reviewer #1: This paper focused on the role of reduced amino acid alphabets in kmer-based disease state prediction using T-cell receptor (TCR) repertoires. Two encoding methods, Atchley factors and BLOSUM62-based distance matrices, are used to cluster amino acids into reduced alphabets. This proposed method is evaluated against standard kmer approaches and existing kmer clustering methods. However, there are several limitations that should be addressed to improve the completeness and robustness of the work.

Major Comments

1. Feature redundancy and model generalizability: The feature importance analysis reveals that only a small fraction of kmer features contribute meaningfully to classification, indicating that most features are redundant. The models may rely heavily on a few high-frequency kmers, potentially limiting its generalization. The authors should consider applying feature selection or dimensionality reduction to mitigate overfitting.

2. Potential information leakage due to sample split strategy: The training and testing sets are split by sample/cohort, but it is unclear whether the sequences across these sets are truly independent. Overlap in sequence regions may result in shared kmers between training and test sets, which could cause inflated performance. The authors should consider alternative data partitioning strategies, such as clustering sequences based on similarity.

Minor Comments

1. Limited model comparison: Only XGBoost is used for classification. A comparison with other traditional classifiers such as Random Forest, Support Vector Machine (SVM), or Logistic Regression would strengthen the analysis and provide insight into whether the performance gains are model-specific.

2. Lack of efficiency analysis: Although the reduced alphabet method is claimed to be more efficient and flexible than existing kmer clustering methods, no empirical or theoretical comparison (e.g., runtime or complexity) is provided. Including such analysis would help substantiate these claims.

3. Representation limitations of kmer features: Compared to deep learning methods (e.g., CNNs or attention-based models), kmer-based representations may suffer from loss of contextual information, limited adaptability, and poor transferability across tasks. The authors should consider discussing the relative strengths and weaknesses of their approach in comparison to more expressive models.

Reviewer #2: The authors present an approach for T-cell receptor (TCR) repertoire classification using amino acid similarity-aware k-mer representations. They used the reduced alphabets for computational efficiency and compared it against standard k-mers and k-mer clustering approaches on cytomegalovirus (CMV) infection and coeliac disease (CeD) datasets. Althouth the work is interesting, several major concerns need to be addressed.

1. The proposed reduced alphabet approach shows only small improvement over standard k-mers. In particular, for CeD classification using test datasets, standard k-mers actually outperform the proposed method (AUROC 0.934 vs 0.860).

The authors should clarify that the proposed method is better than the basic k-mer method and others.

2. Furthermore, it would helpful to provide runtime comparisons between your proposed method and others to support the computational efficiency.

3. The authors used XGBoost with a fixed hyperparameter for the classifiction. How did the hyperparameters are chosen? They should present how they found the optimal hyperparameters in details.

4. In addition, the authors should show and compare the experimental results with other classfiers (Random Forests, SVMs, Artificial Neural Networks, etc.).

5. Why did the authors downsample the datasets at the experiments? Even though they already provided some explanation, a more detailed and comprehensive discussion is needed for clarity. If possible, please show the experimental results without the downsampling to clarify the impact of the decision.

6. The XGBoost model uses thousands of features, making interpretation challenging. Although the authors performed the feature importance analyses (Figures 3 and 6, S1-S2), it provides little biological understanding. The authors should include some biological insights clearly. For example, they should show clearly whether the identified motifs have some direct relationships to the previously known TCR-antigen interactions.

7. The sample size of CeD datasets is limited. Is the number of samples sufficient to train the model effectively? In addition, the CeD datasets exhibit a significant imbalance between the positive and negative samples than CMV datasets. The authors should provide a clear and detailed discussion on why both the small number of samples and the imbalanced datasets were not significant problems if they claim this to be the case.

Reviewer #3: The immune system keeps a record of past infections and current immune state. Advances in sequencing now allow large-scale profiling of the adaptive immune receptor repertoire, which encodes this record. However, linking immune repertoires to functional outcomes remains a major challenge. Machine learning offers a promising path forward, yet the immense diversity of immune receptors often necessitates some form of coarse-graining of sequence information before model training.

This paper evaluates the use of kmers derived from reduced amino acid alphabets as coarse-grained representations of immune repertoires. Kmer-based classification remains a valuable baseline approach, even in the era of deep learning. Extending this method through reduced amino acid alphabets is an interesting idea with potential to advance the field. However, the current manuscript has several important limitations.

Major concerns:

1. The performance improvements obtained using reduced amino acid alphabets are not statistically significant in cross-validation. For instance, AUROC scores in CMV classification (Table 2: 0.75 ± 0.03 vs. 0.77 ± 0.04) and CeD classification (Table 5) show no clear evidence that reduced alphabet representations outperform baseline. Similarly, there is no consistent advantage of BLOSUM-based alphabets over Atchley-based ones. Can the authors identify a specific scenario where reduced alphabets yield a meaningful benefit? If not, the abstract and discussion should be revised to more accurately reflect the evidence provided.

2. All reported models perform substantially worse on the CMV dataset than the method introudced by Emerson et al. (Nature Genetics 2017, doi:10.1038/ng.3822), who first described this dataset. The manuscript should address this discrepancy directly and more clearly contextualize the contribution of the present work relative to methods with demonstrated higher performance.

Minor concerns:

1. The introduction states that "the underlying relationships between amino acids that may lead to shared function are unknown". However, recent works have begun to address this gap, for example Henderson et al. PNAS 2024 (doi:10.1073/pnas.2408696121) compared the ability of different reduced amino acid alphabets to capture functional similarity using known TCR–pMHC pairings.

2. On lines 119-124, does the similarity threshold refers to the hierarchical clustering of Atchley factors or BLOSUM62? At each similarity threshold a fixed integer-sized reduced alphabet is obtained, so it was unclear what is meant with the effective number of 3.16.

**Have the authors made all data and (if applicable) computational code underlying the findings in their manuscript fully available?**

The PLOS Data policy requires authors to make all data and code underlying the findings described in their manuscript fully available without restriction, with rare exception (please refer to the Data Availability Statement in the manuscript PDF file). The data and code should be provided as part of the manuscript or its supporting information, or deposited to a public repository. For example, in addition to summary statistics, the data points behind means, medians and variance measures should be available. If there are restrictions on publicly sharing data or code —e.g. participant privacy or use of data from a third party—those must be specified.requires authors to make all data and code underlying the findings described in their manuscript fully available without restriction, with rare exception (please refer to the Data Availability Statement in the manuscript PDF file). The data and code should be provided as part of the manuscript or its supporting information, or deposited to a public repository. For example, in addition to summary statistics, the data points behind means, medians and variance measures should be available. If there are restrictions on publicly sharing data or code —e.g. participant privacy or use of data from a third party—those must be specified.requires authors to make all data and code underlying the findings described in their manuscript fully available without restriction, with rare exception (please refer to the Data Availability Statement in the manuscript PDF file). The data and code should be provided as part of the manuscript or its supporting information, or deposited to a public repository. For example, in addition to summary statistics, the data points behind means, medians and variance measures should be available. If there are restrictions on publicly sharing data or code —e.g. participant privacy or use of data from a third party—those must be specified.requires authors to make all data and code underlying the findings described in their manuscript fully available without restriction, with rare exception (please refer to the Data Availability Statement in the manuscript PDF file). The data and code should be provided as part of the manuscript or its supporting information, or deposited to a public repository. For example, in addition to summary statistics, the data points behind means, medians and variance measures should be available. If there are restrictions on publicly sharing data or code —e.g. participant privacy or use of data from a third party—those must be specified.

Reviewer #1: Yes

Reviewer #2: None

Reviewer #3: Yes

PLOS authors have the option to publish the peer review history of their article (what does this mean?). If published, this will include your full peer review and any attached files.). If published, this will include your full peer review and any attached files.). If published, this will include your full peer review and any attached files.). If published, this will include your full peer review and any attached files.

...

Reviewer #1: No

Reviewer #2: No

Reviewer #3: No

**Figure resubmission:**
---

## [Decision Letter · Decision Letter 1]

16 Dec 2025

PCOMPBIOL-D-25-01274R1

Evaluating the utility of amino acid similarity-aware kmers to represent TCR repertoires for classification

PLOS Computational Biology

Dear Dr. Fowler,

Thank you for submitting your manuscript to PLOS Computational Biology. After careful consideration, we feel that it has merit but does not fully meet PLOS Computational Biology's publication criteria as it currently stands. Therefore, we invite you to submit a revised version of the manuscript that addresses the points raised during the review process.

We look forward to receiving your revised manuscript.

Kind regards,

Shanfeng Zhu, Ph.D.

Academic Editor

PLOS Computational Biology

Mark Tanaka

Section Editor

PLOS Computational Biology

**Additional Editor Comments:**

The second reviewer recommends conducting hyperparameter tuning to strengthen the study's methodological rigor.

**Journal Requirements:**

1) We notice that your supplementary Tables are included in the manuscript file. Please remove them and upload them with the file type 'Supporting Information'. Please ensure that each Supporting Information file has a legend listed in the manuscript after the references list.

**Reviewers' comments:**

Reviewer's Responses to Questions

**Comments to the Authors:**

Reviewer #1: This paper focused on the role of reduced amino acid alphabets in kmer-based disease state prediction using T-cell receptor (TCR) repertoires. Two encoding methods, Atchley factors and BLOSUM62-based distance matrices, are used to cluster amino acids into reduced alphabets. This proposed method is evaluated against standard kmer approaches and existing kmer clustering methods.

The authors have made substantial improvements through revisions, addressing both major and minor concerns from earlier reviews. However, they should remain mindful of the importance of maintaining the software/web tool, as we have observed cases where code or model become inaccessible soon after publication.

To conclude, I would recommend moving forward with the publication of this paper, as the necessary revisions have been made, and it is ready for submission in its final form.

Reviewer #2: The authors have adequately addressed most of the previous concerns. However, regarding comment #3, the justification for using fixed hyperparameters remains unclear. While the authors provided additional explanation in their response, I recommend that the authors conduct hyperparameter tuning to identify optimal parameters, which would strengthen the methodological rigor of the study.

Reviewer #3: The authors have satisfactorily addressed my concerns and questions.

**Have the authors made all data and (if applicable) computational code underlying the findings in their manuscript fully available?**

The PLOS Data policy requires authors to make all data and code underlying the findings described in their manuscript fully available without restriction, with rare exception (please refer to the Data Availability Statement in the manuscript PDF file). The data and code should be provided as part of the manuscript or its supporting information, or deposited to a public repository. For example, in addition to summary statistics, the data points behind means, medians and variance measures should be available. If there are restrictions on publicly sharing data or code —e.g. participant privacy or use of data from a third party—those must be specified.requires authors to make all data and code underlying the findings described in their manuscript fully available without restriction, with rare exception (please refer to the Data Availability Statement in the manuscript PDF file). The data and code should be provided as part of the manuscript or its supporting information, or deposited to a public repository. For example, in addition to summary statistics, the data points behind means, medians and variance measures should be available. If there are restrictions on publicly sharing data or code —e.g. participant privacy or use of data from a third party—those must be specified.requires authors to make all data and code underlying the findings described in their manuscript fully available without restriction, with rare exception (please refer to the Data Availability Statement in the manuscript PDF file). The data and code should be provided as part of the manuscript or its supporting information, or deposited to a public repository. For example, in addition to summary statistics, the data points behind means, medians and variance measures should be available. If there are restrictions on publicly sharing data or code —e.g. participant privacy or use of data from a third party—those must be specified.requires authors to make all data and code underlying the findings described in their manuscript fully available without restriction, with rare exception (please refer to the Data Availability Statement in the manuscript PDF file). The data and code should be provided as part of the manuscript or its supporting information, or deposited to a public repository. For example, in addition to summary statistics, the data points behind means, medians and variance measures should be available. If there are restrictions on publicly sharing data or code —e.g. participant privacy or use of data from a third party—those must be specified.

Reviewer #1: None

Reviewer #2: None

Reviewer #3: Yes

PLOS authors have the option to publish the peer review history of their article (what does this mean?). If published, this will include your full peer review and any attached files.). If published, this will include your full peer review and any attached files.). If published, this will include your full peer review and any attached files.). If published, this will include your full peer review and any attached files.

...

Reviewer #1: No

Reviewer #2: No

Reviewer #3: No

**Figure resubmission:**
---

## [Decision Letter · Decision Letter 2]

2 Mar 2026

PCOMPBIOL-D-25-01274R2

Evaluating the utility of amino acid similarity-aware kmers to represent TCR repertoires for classification

PLOS Computational Biology

Dear Dr. Fowler,

Thank you for submitting your manuscript to PLOS Computational Biology. After careful consideration, we feel that it has merit but does not fully meet PLOS Computational Biology's publication criteria as it currently stands. Therefore, we invite you to submit a revised version of the manuscript that addresses the points raised during the review process.

We look forward to receiving your revised manuscript.

Kind regards,

Shanfeng Zhu, Ph.D.

Academic Editor

PLOS Computational Biology

Mark Tanaka

Section Editor

PLOS Computational Biology

**Additional Editor Comments:**

The reviewer is basically satisfied with the revision, with some minor issues on the github repository. Please address these issues.

**Reviewers' comments:**

Reviewer's Responses to Questions

**Comments to the Authors:**

Reviewer #2: The authors have fully addressed the previous concerns. The revised manuscript, along with the updated tables and figures, comprehensively reflected these improvements.

The only remaining one is the computational reproducibility and code availability via the provided GitHub repository (https://github.com/hannrko/enc_kmer_tcr_models).

- The authors should ensure that the newly developed scripts (Bayesian optimization (Optuna), the 80/20 train-validation fold splitting, etc.) are successfully committed to the repository.

- Please include the README.md file to provide clear, step-by-step instructions on how to run the newly tuned models.

- Also, the authors should confirm that the requirements.txt or environment.yml file is up-to-date. It would include the specific versions of the core libraries.

**Have the authors made all data and (if applicable) computational code underlying the findings in their manuscript fully available?**

The PLOS Data policy requires authors to make all data and code underlying the findings described in their manuscript fully available without restriction, with rare exception (please refer to the Data Availability Statement in the manuscript PDF file). The data and code should be provided as part of the manuscript or its supporting information, or deposited to a public repository. For example, in addition to summary statistics, the data points behind means, medians and variance measures should be available. If there are restrictions on publicly sharing data or code —e.g. participant privacy or use of data from a third party—those must be specified.requires authors to make all data and code underlying the findings described in their manuscript fully available without restriction, with rare exception (please refer to the Data Availability Statement in the manuscript PDF file). The data and code should be provided as part of the manuscript or its supporting information, or deposited to a public repository. For example, in addition to summary statistics, the data points behind means, medians and variance measures should be available. If there are restrictions on publicly sharing data or code —e.g. participant privacy or use of data from a third party—those must be specified.requires authors to make all data and code underlying the findings described in their manuscript fully available without restriction, with rare exception (please refer to the Data Availability Statement in the manuscript PDF file). The data and code should be provided as part of the manuscript or its supporting information, or deposited to a public repository. For example, in addition to summary statistics, the data points behind means, medians and variance measures should be available. If there are restrictions on publicly sharing data or code —e.g. participant privacy or use of data from a third party—those must be specified.requires authors to make all data and code underlying the findings described in their manuscript fully available without restriction, with rare exception (please refer to the Data Availability Statement in the manuscript PDF file). The data and code should be provided as part of the manuscript or its supporting information, or deposited to a public repository. For example, in addition to summary statistics, the data points behind means, medians and variance measures should be available. If there are restrictions on publicly sharing data or code —e.g. participant privacy or use of data from a third party—those must be specified.

Reviewer #2: None

PLOS authors have the option to publish the peer review history of their article (what does this mean?). If published, this will include your full peer review and any attached files.). If published, this will include your full peer review and any attached files.). If published, this will include your full peer review and any attached files.). If published, this will include your full peer review and any attached files.

...

Reviewer #2: No

**Figure resubmission:**
---

## [Decision Letter · Decision Letter 3]

7 Apr 2026

Dear Dr. Fowler,

We are pleased to inform you that your manuscript 'Evaluating the utility of amino acid similarity-aware kmers to represent TCR repertoires for classification' has been provisionally accepted for publication in PLOS Computational Biology.

Best regards,

Shanfeng Zhu, Ph.D.

Academic Editor

PLOS Computational Biology

Mark Tanaka

Section Editor

PLOS Computational Biology

All reviewers are satisfied with the revisions.

Reviewer's Responses to Questions

**Comments to the Authors:**

Reviewer #2: I do not have any further comments.

**Have the authors made all data and (if applicable) computational code underlying the findings in their manuscript fully available?**

The PLOS Data policy requires authors to make all data and code underlying the findings described in their manuscript fully available without restriction, with rare exception (please refer to the Data Availability Statement in the manuscript PDF file). The data and code should be provided as part of the manuscript or its supporting information, or deposited to a public repository. For example, in addition to summary statistics, the data points behind means, medians and variance measures should be available. If there are restrictions on publicly sharing data or code —e.g. participant privacy or use of data from a third party—those must be specified.requires authors to make all data and code underlying the findings described in their manuscript fully available without restriction, with rare exception (please refer to the Data Availability Statement in the manuscript PDF file). The data and code should be provided as part of the manuscript or its supporting information, or deposited to a public repository. For example, in addition to summary statistics, the data points behind means, medians and variance measures should be available. If there are restrictions on publicly sharing data or code —e.g. participant privacy or use of data from a third party—those must be specified.requires authors to make all data and code underlying the findings described in their manuscript fully available without restriction, with rare exception (please refer to the Data Availability Statement in the manuscript PDF file). The data and code should be provided as part of the manuscript or its supporting information, or deposited to a public repository. For example, in addition to summary statistics, the data points behind means, medians and variance measures should be available. If there are restrictions on publicly sharing data or code —e.g. participant privacy or use of data from a third party—those must be specified.requires authors to make all data and code underlying the findings described in their manuscript fully available without restriction, with rare exception (please refer to the Data Availability Statement in the manuscript PDF file). The data and code should be provided as part of the manuscript or its supporting information, or deposited to a public repository. For example, in addition to summary statistics, the data points behind means, medians and variance measures should be available. If there are restrictions on publicly sharing data or code —e.g. participant privacy or use of data from a third party—those must be specified.

Reviewer #2: None

PLOS authors have the option to publish the peer review history of their article (what does this mean?). If published, this will include your full peer review and any attached files.). If published, this will include your full peer review and any attached files.). If published, this will include your full peer review and any attached files.). If published, this will include your full peer review and any attached files.

...

Reviewer #2: No

---

## [Editor Report · Acceptance letter]

PCOMPBIOL-D-25-01274R3

Evaluating the utility of amino acid similarity-aware kmers to represent TCR repertoires for classification

Dear Dr Fowler,

I am pleased to inform you that your manuscript has been formally accepted for publication in PLOS Computational Biology. Your manuscript is now with our production department and you will be notified of the publication date in due course.

With kind regards,

Aiswarya Satheesan
